# Arctic observations and numerical simulations of surface wind effects on Multi-Angle Snowflake Camera measurements

Kyle E. Fitch[1,2], Chaoxun Hang[3,4], Ahmad Talaei[1], and Timothy J. Garrett[1]

[1]Department of Atmospheric Sciences, University of Utah, Salt Lake City, 84112, USA
[2]Department of Engineering Physics, Air Force Institute of Technology, Wright-Patterson Air Force Base, Ohio, 45433, USA
[3]Department of Civil Engineering, Monash University, Clayton, 3168, Australia
[4]School of Oceanography, Shanghai Jiao Tong University, Shanghai, 200240, China

**Correspondence:** Kyle Fitch (kyle.fitch@afit.edu)

**Abstract.**

Ground-based measurements of frozen precipitation are heavily influenced by interactions of surface winds with gauge-shield geometry. The Multi-Angle Snowflake Camera (MASC), which photographs hydrometeors in free-fall from three different angles while simultaneously measuring their fall speed, has been used in the field at multiple mid-latitude and polar locations both with and without wind shielding. Here we present an analysis of Arctic field observations — with and without a Belfort double Alter shield — and compare the results to computational fluid dynamics (CFD) simulations of the airflow and corresponding particle trajectories around the unshielded MASC. MASC-measured fall speeds compare well with Ka-band Atmospheric Radiation Measurement (ARM) Zenith Radar (KAZR) mean Doppler velocities only when winds are light ($\leq 5\,\mathrm{m\,s^{-1}}$) and the MASC is shielded. MASC-measured fall speeds that do not match KAZR measured velocities tend to fall below a threshold value that increases approximately linearly with wind speed but is generally $< 0.5\,\mathrm{m\,s^{-1}}$. For those events with wind speeds $\leq 1.5\,\mathrm{m\,s^{-1}}$, hydrometeors fall with an orientation angle mode of $12°$ from the horizontal plane, and large, low-density aggregates are as much as five times more likely to be observed. Simulations in the absence of a wind shield show a separation of flow at the upstream side of the instrument, with an upward velocity component just above the aperture, which decreases the mean particle fall speed by 55% (74%) for a wind speed of $5\,\mathrm{m\,s^{-1}}$ ($10\,\mathrm{m\,s^{-1}}$). We conclude that accurate MASC observations of the microphysical, orientation, and fall speed characteristics of snow particles require shielding by a double wind fence and restriction of analysis to events where winds are light ($\leq 5\,\mathrm{m\,s^{-1}}$). Hydrometeors do not generally fall in still air, so adjustments to these properties' distributions within natural turbulence remain to be determined.

## 1 Introduction

Accurate measurement of snowfall is of importance to a wide range of scientific and public interests, including weather and climate prediction and monitoring (Yang et al., 2005; Rasmussen et al., 2012; Thériault et al., 2015; Mekis et al., 2018), hydrological cycles (Yang et al., 2005; Rasmussen et al., 2012; Thériault et al., 2012; Mekis et al., 2018), ecosystem research (Rasmussen et al., 2012), snowpack monitoring and disaster management (Thériault et al., 2015; Mekis et al., 2018), trans-

portation (Rasmussen et al., 2001; Thériault et al., 2012, 2015), agriculture (Mekis et al., 2018), and resource management (Thériault et al., 2015; Mekis et al., 2018).

A persistent limitation of these studies is that catch-style precipitation gauges are prone to large uncertainties, especially when measuring snowfall in high winds — a bias referred to as "under-catch" (Groisman et al., 1991; Groisman and Legates, 1994; Goodison et al., 1998; Rasmussen et al., 2001; Yang et al., 2005). A common remedy is to apply a correction based primarily on wind speed (Yang et al., 1993; Rasmussen et al., 2001, 2012; Wolff et al., 2015), although hydrometeor type (Thériault et al., 2012) and a dynamic drag coefficient (Colli et al., 2015) may also be considered. The correction is calculated

by measuring the collection efficiency for a particular gauge or gauge-shield geometry, where collection efficiency is defined as the ratio of the gauge-measured precipitation rate to the best-estimate rate (Thériault et al., 2012). The Double Fence Intercomparison Reference (DFIR) is the standard reference, as determined by the World Meteorological Organization (WMO; Goodison et al., 1998); however, the DFIR has its own uncertainties which can lead to underestimation (Yang et al., 1993) or even overestimation (Thériault et al., 2015) of snowfall rates.

Surface-based measurements of solid precipitation fall speed (Garrett and Yuter, 2014), fall orientation (Garrett et al., 2015; Jiang et al., 2019), and size distributions (Thériault et al., 2012) are all very sensitive to wind speed, with fall speed and size distribution having a strong influence on precipitation gauge collection efficiency (Thériault et al., 2012, 2015). Accurate measurement of solid precipitation characteristics is important for constraining the densities and size distributions used in bulk microphysical parameterizations (e.g., Thompson et al., 2008; Morrison and Milbrandt, 2015). These parameters strongly

influence bulk fall speed, highlighted by the Intergovernmental Panel on Climate Change (IPCC) as a critical factor for determining climate sensitivity (Flato et al., 2013). Likewise, knowledge of preferential hydrometeor orientation angles leads to the improved inference of hydrometeor shapes from backscattered polarimetric radar intensities (Vivekanandan et al., 1991, 1994; Matrosov et al., 2005; Matrosov, 2015), and these shapes combine with density to determine hydrometeor fall speeds (Böhm, 1989).

Past studies have typically combined airflow modeling and field observations to better understand the measurement error induced by winds and gauge geometry. Computational fluid dynamics (CFD) calculations are used to characterize the wind velocity field and its interaction with various stationary objects in turbulent flows (Moat et al., 2006; Dehbi, 2008; Ferrari et al., 2017). Thériault et al. (2012) combined field observations and CFD simulations to better understand the scatter in collection efficiency as a function of wind speed for a Geonor, Inc. precipitation gauge located in a single Alter shield. Findings

suggested that in addition to wind speed, the hydrometeor collection efficiency is a function of both hydrometeor type and size distribution. For example, hydrometeors such as graupel, with a relatively large density-to-surface-area ratio, fall faster and are collected more efficiently than large, low-density, aggregate-type hydrometeors. Additionally, Colli et al. (2016a, b) compared shielded and unshielded gauge configurations using both time-averaged and time-dependent CFD simulations and found that a single Alter shield was effective in reducing the magnitude of turbulent flow above the gauge aperture. However, upwind shield

deflector fins still produced turbulence that propagated into the collection area and generally reduced the collection efficiency. CFD simulations were also analyzed for wind flow along the optical axis of a snowflake video imager, with eddies dissipating

approximately 1 m downstream of the camera housing, resulting in only minor modifications to the wind field (Newman et al., 2009).

One instrument that has received increased attention, but whose sampling characteristics have yet to be characterized in detail, is the Multi-Angle Snowflake Camera (MASC; Garrett et al., 2012). The MASC system has overall dimensions of 43.5 cm x 58 cm x 21.5 cm (Stuefer and Bailey, 2016) and observes particles falling into a hollow decagonal-prism-shaped collection volume. The system's casing houses three cameras focused on a point at the center of the collection volume 10 cm away, with each camera separated by 36° (for more details, see Fig. 1 from Garrett et al., 2012). A coupled system of directly opposing near-infrared emitters and detectors, vertically separated by 32 mm, detects falling hydrometeors larger than ~0.1 mm in maximum dimension (Garrett and Yuter, 2014). This triggers the cameras and three high-powered LEDs located directly above on top of the casing. The time between triggers of the upper and lower emitter-detector pairs yields a fall speed. High-resolution images are captured at an exposure time of 1/25,000th of a second, sufficient to capture a vertical resolution of 40 μm in a hydrometeor falling at $1\,\mathrm{m\,s^{-1}}$ (Garrett et al., 2012).

The MASC system has helped to advance precipitation measurement by automating simultaneous high-resolution photography and fall speed measurement of falling hydrometeors from multiple angles, removing the need for tedious manual collection. Variables derived from the high-resolution images include those describing a hydrometeor's size, shape, fall orientation, and approximate riming degree (Garrett et al., 2012; Garrett and Yuter, 2014; Garrett et al., 2015). As these hydrometeor properties are crucial for accurate numerical modeling and microwave scattering calculations, the MASC has been used at various polar and mid-latitude locations to constrain microphysical characteristics (e.g., Grazioli et al., 2017; Dunnavan et al., 2019; Jiang et al., 2019; Vignon et al., 2019), improve radar-based estimates of snowfall rates (Gergely and Garrett, 2016; Cooper et al., 2017; Schirle et al., 2019), automatically classify hydrometeors (Praz et al., 2017; Besic et al., 2018; Hicks and Notaroš, 2019; Leinonen and Berne, 2020; Schaer et al., 2020), reconstruct particle shapes (Notaroš et al., 2016; Kleinkort et al., 2017) and size distributions (Cooper et al., 2017; Huang et al., 2017; Schirle et al., 2019), and as ground truth comparisons for radar measurements (Bringi et al., 2017; Gergely et al., 2017; Matrosov et al., 2017; Kennedy et al., 2018; Oue et al., 2018; Matrosov et al., 2019). Unlike more common precipitation gauges, the wind velocity field in the proximity of the MASC has not been simulated for various surface winds speeds, directions, or turbulence kinetic energies (TKE).

Studies of hydrometeor behaviors using the MASC have shown, somewhat surprisingly, that frozen hydrometeor fall speeds are only weakly dependent on their size or shape, particularly under conditions of high turbulence intensity (Garrett and Yuter, 2014). Prior studies had shown a much stronger dependence but had theoretically assumed or experimentally arranged for falling hydrometeors to settle in still air (Locatelli and Hobbs, 1974; Böhm, 1989). MASC measurements led to a hypothesis that snow "swirls" in turbulent air in a manner that spreads particle fall speeds to both higher and lower values (Garrett and Yuter, 2014) — an effect shown in prior work to be non-negligible in turbulent flows (Nielsen, 2007). While the fact that snowflakes can just as readily move upwards as downwards is easily verified by any casual observations of a winter storm, it has remained unclear the extent to which the measurements of snowflake fall speed obtained by the MASC have been reflective of reality rather than some artifact of interactions of surrounding winds with the instrument body.

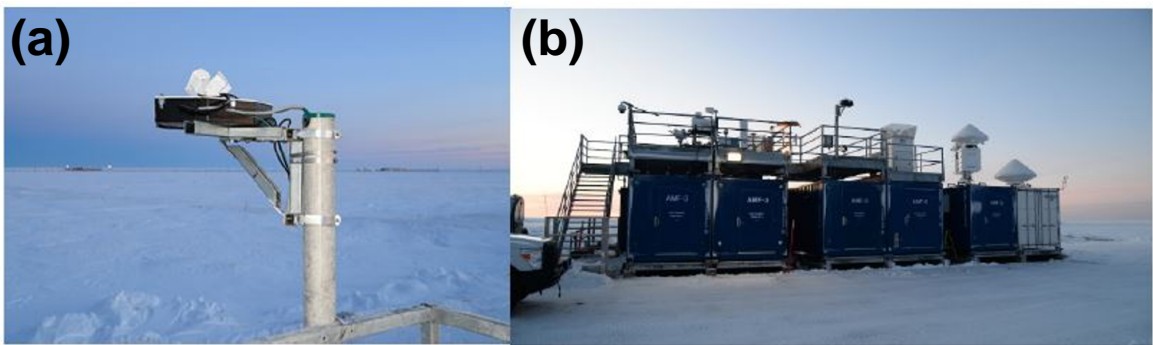

**Figure 1.** (a) Unshielded MASC configuration at the Third ARM Mobile Facility (AMF3), Oliktok Point, Alaska. (b) Ground-level view of the MASC on top of a group of shipping containers. This was the MASC configuration from initial deployment in February 2015 through 21 August 2016. Image courtesy of the U.S. Department of Energy Atmospheric Radiation Measurement (ARM) user facility.

In this study, we analyze field observations of hydrometeor characteristics from a MASC located in the Arctic and compare these results to CFD simulations of hydrometeor-MASC interactions. The goal of this study is to better understand and characterize the influence of ambient wind speeds on MASC measurements of hydrometeor fall speed, fall orientation, size distribution, and riming degree for both wind-shielded and unshielded configurations.

## 2 Hydrometeor observations

### 2.1 Methods

Processing of MASC imagery consists of distinguishing foreground pixels from background to define the region of interest (ROI) and then fitting the ROI with a bounding ellipse (Shkurko et al., 2018). The maximum dimension $D_{max}$ is defined as the length of the ellipse's major axis for each image. The absolute value of the angle between the major axis and the local horizontal plane is the orientation angle $\theta$ (Garrett et al., 2012; Garrett and Yuter, 2014; Garrett et al., 2015; Shkurko et al., 2018). A complexity parameter $\chi$ is used to distinguish riming classes (Garrett and Yuter, 2014). Here we use $\chi \leq 1.35$ to identify heavily rimed graupel, $1.35 < \chi \leq 2.00$ for moderate riming, and $\chi > 2.00$ indicates sparsely-rimed aggregates. We note that a value of 1.75 was used to distinguish moderately rimed particles from aggregates for Utah snow measurements in Garrett and Yuter (2014), with the observation that the value is subjectively determined by visual inspection of hydrometeor images and varies with location. Mean values of fall speed $v_p$, $D_{max}$, $\theta$, and $\chi$ from all three images are used for each particle.

A MASC was installed at the Department of Energy's Third Atmospheric Radiation Measurement (ARM) Mobile Facility (AMF3), Oliktok Point, Alaska, in February 2015. The initial deployment was atop a group of shipping containers with no wind shield (Fig. 1). On 22 August 2016, the MASC was relocated to ground level and placed inside of a Belfort Model 36001 Double Alter Wind Shield (Fig. 2). The central camera was pointed in the east-northeasterly direction (Jiang et al., 2019), with

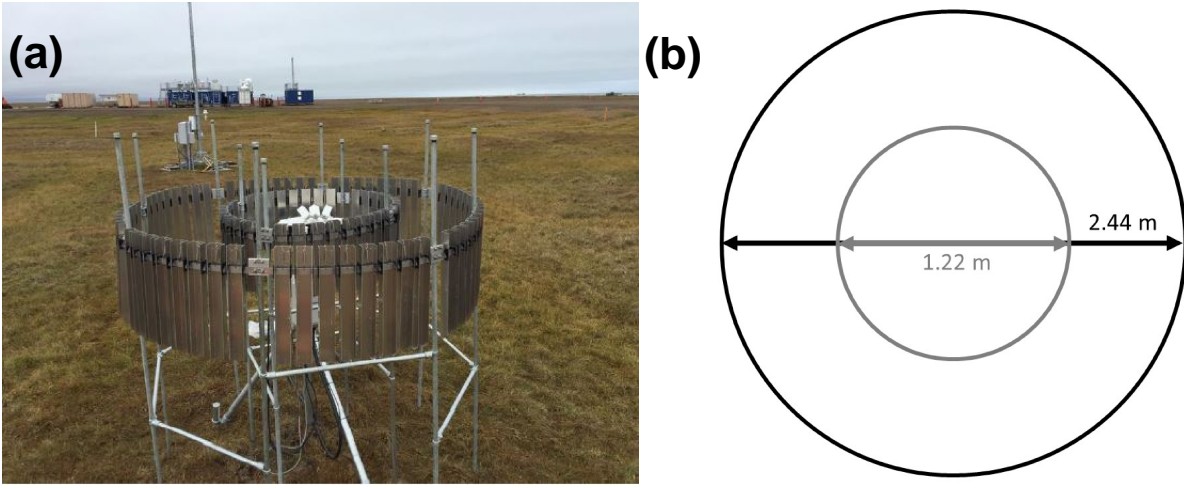

**Figure 2.** (a) The MASC was relocated to ground level and placed inside a Belfort double Alter shield on 22 August 2016 (field site photograph courtesy of Martin Stuefer). (b) The shield consists of inner and outer fences with diameters of 1.22 m and 2.44 m, respectively.

surface wind observations showing this to be the predominant wind direction for the present study. The inner (outer) fence of the shield is 1.22 (2.44) m in diameter, with 32 (64) deflector fins that are each 46 (61) cm in length. Observations used here include both unshielded and shielded configurations, spanning a 33-month period from 29 November 2015 to 28 August 2018 (ARM Climate Research Facility, 2014). Raw data and images were processed with a local University of Utah processing suite called *mascpy* (The Hive: University of Utah Research Data Repository, 2020a, b), similar to that described in Shkurko et al. (2018).

A total of 158,057 particles from 266 distinct events are included here for analysis, with 51 events from the unshielded period of 29 November 2015 to 21 August 2016, and 215 events from the shielded period of 22 August 2016 to 28 August 2018. Distinct events were identified by a length of time between MASC precipitation measurements of $> 12$ hours, or by a length of time of $> 3$ hours with an accompanying change of pressure of at least 2 mb. These thresholds were determined by analyzing the period of 4 to 17 December 2017, during which 14,528 precipitation particles were associated with five distinct events as determined by manual inspection of the KAZR reflectivity time series (not shown). Differences in riming class composition for various wind speed categories are determined to be statistically significant by comparing $\chi$ distributions using the two-sample Kolmogorov-Smirnov Test at a 5% significance level. In each test, one sample is from the high-wind category ($U_{sfc} > 5\,\mathrm{m\,s}^{-1}$) and the other is from the respective low-wind category.

To complement MASC observations and characterize the influence of ambient wind speed on MASC measurements, surface wind measurements from a traditional meteorological ground suite (Ritsche, 2011; ARM Climate Research Facility, 2013) were matched to MASC hydrometeors by calculating a mean wind speed $U_{sfc}$ for the 1 minute period leading up to the observation time corresponding to each hydrometeor. The wind measurement is taken at a standard height of 10 m, which is estimated to be 5 (9) m higher than the unshielded (shielded) MASC shown in Fig. 1 (2). In addition to the quality control checks listed in Shkurko et al. (2018), a surface temperature threshold of $< 2\,^{\circ}\mathrm{C}$ was used to exclude liquid hydrometeors, which are occasionally misidentified by the *mascpy* algorithm.

For comparison to MASC fall speeds, mean Doppler velocity was calculated from the volume of scattering hydrometeors detected by a co-located Ka-band ARM Zenith-pointing Radar (KAZR). At a vertical resolution of 30 m, the KAZR produces measurements of the first three moments of the Doppler spectrum: reflectivity, mean Doppler velocity, and spectrum width (Widener et al., 2012; Oue et al., 2018). The Doppler velocity signal has a resolution of $0.05\ \mathrm{m\,s^{-1}}$ (Oue et al., 2018) and consists of both larger particle fall speeds and the vertical air motions traced by smaller particles (Shupe et al., 2008). Using only Doppler velocity measurements originating from below cloud base, we isolate the signal of the larger, precipitation-sized hydrometeors. Both mean Doppler velocity and cloud base height were retrieved from the ARM's KAZR Active Remote Sensing of CLouds (ARSCL) Value-Added Product (ARM Climate Research Facility, 2015; Clothiaux et al., 2000).

Results are presented here in the form of probability density function (PDF) estimates, calculated using a kernel density estimator of the form

$$\hat{f}(x_0) = \frac{1}{n_s h} \sum_{i=1}^{n_s} K\left(\frac{x_0 - x_i}{h}\right) \tag{1}$$

where $x_0$ is a real value of the distribution being estimated, $x_i$ is a random sample from the distribution, $n_s$ is the sample size, and $h$ is the bandwidth (Wilks, 2011). The Gaussian smoothing function is $K(x) = (2\pi)^{-1/2}\exp(-x^2/2)$ for a random variable $x$, and $h$ is optimized according to Bowman and Azzalini (1997) to produce a smooth curve. For distributions of $D_{max}$, the exponential slope parameter $\lambda$ is computed using a linear least squares regression from the peak of the log-linear distribution through the tail.

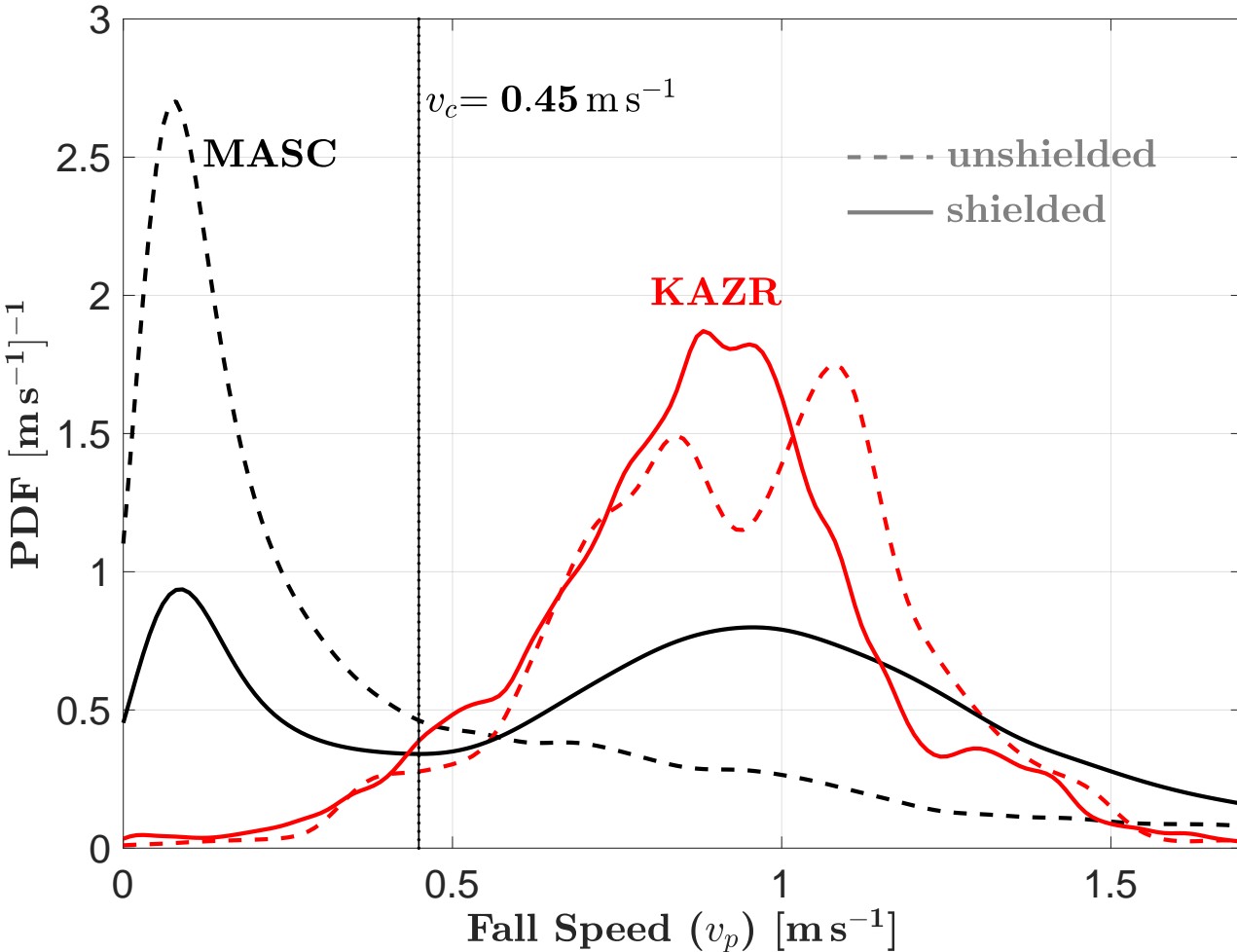

**Figure 3.** Comparison of fall speed $v_p$ probability density function (PDF) estimates from MASC and KAZR measurements, both with and without wind shielding of the MASC. KAZR measurements are from the mean Doppler velocity below cloud base (positive downward, see Sect. 2.1 for details). The cutoff fall speed $v_c$ marks the location of the local minimum separating the two modes of the shielded MASC distribution. Unshielded and shielded MASC observations are from two separate periods: 29 November 2015 to 21 August 2016 and 22 August 2016 to 28 August 2018, respectively.

## 2.2 Observations of fall speed

Distributions of MASC-measured particle fall speed $v_p$, both with and without a wind shield, are compared to coincident measurements from the KAZR in Fig. 3. The KAZR mean Doppler velocity mode is $\sim 1\,\mathrm{m\,s^{-1}}$, while the MASC-measured fall speed distribution has a mode of $0.08\,\mathrm{m\,s^{-1}}$ for both the shielded and unshielded cases. However, the shielded MASC fall

speed distribution has a second mode at $0.96 \text{ m s}^{-1}$, similar to the location of the KAZR mode. Notably, a low-speed mode was not observed in the KAZR measurements despite its velocity resolution of $0.05 \text{ m s}^{-1}$.

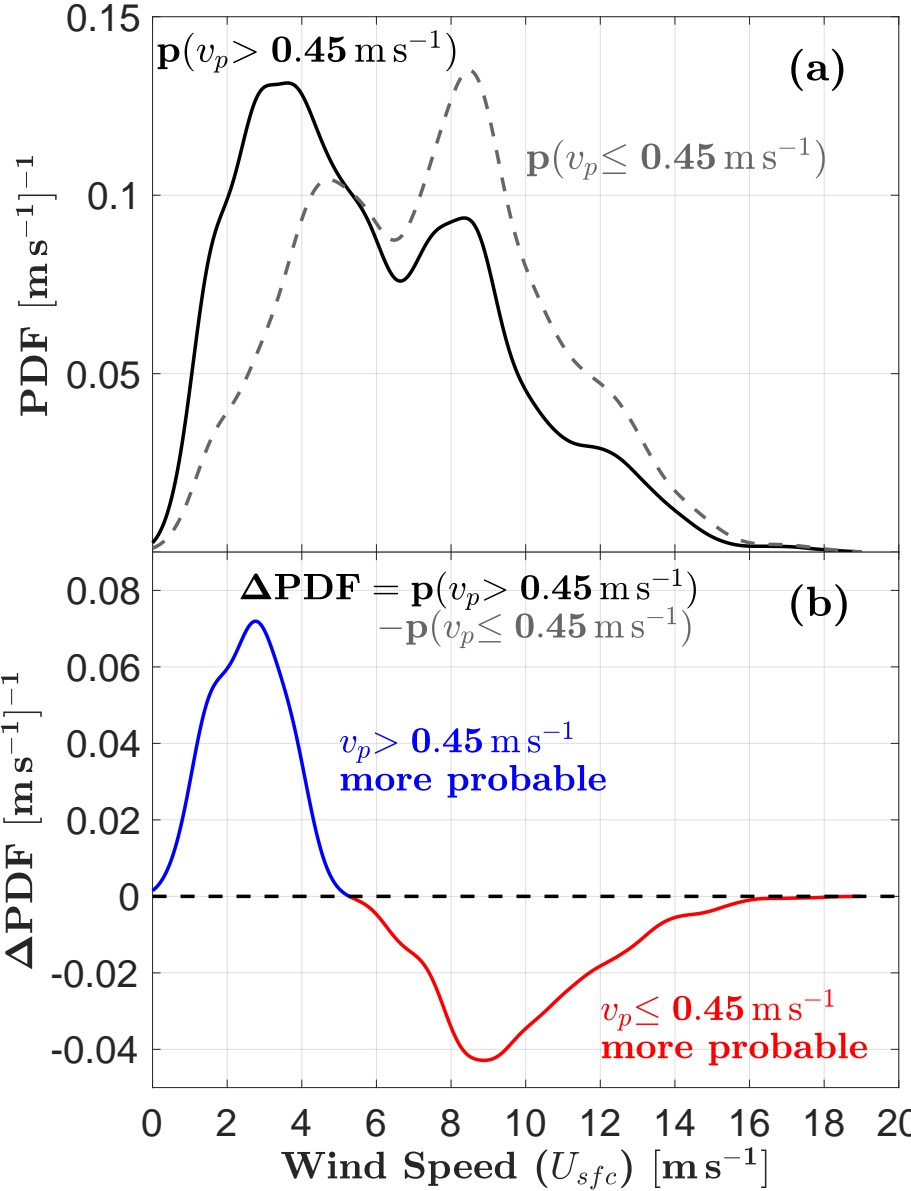

**Figure 4.** (a) Comparison of, and (b) difference between, estimates of surface wind speed $U_{sfc}$ PDFs for the high ($v_p > 0.45\,\mathrm{m\,s^{-1}}$) and low ($v_p \leq 0.45\,\mathrm{m\,s^{-1}}$) fall speed modes of the shielded MASC fall speed distribution from Fig. 3. $\Delta PDF > 0$ means the probability of $v_p > 0.45\,\mathrm{m\,s^{-1}}$ is greater.

The shielded MASC fall speed distribution deviates substantially from the corresponding KAZR distribution for fall speeds below $0.45\,\mathrm{m\,s^{-1}}$. This is the location of the local minimum separating the two modes of the shielded MASC fall speed distribution and is defined from here on as the cutoff fall speed $v_c$: the fall speed below which MASC measurements are assumed to be erroneous. The fall speed distribution can therefore be divided into two parts: $v_p > v_c$ and $v_p \leq v_c$.

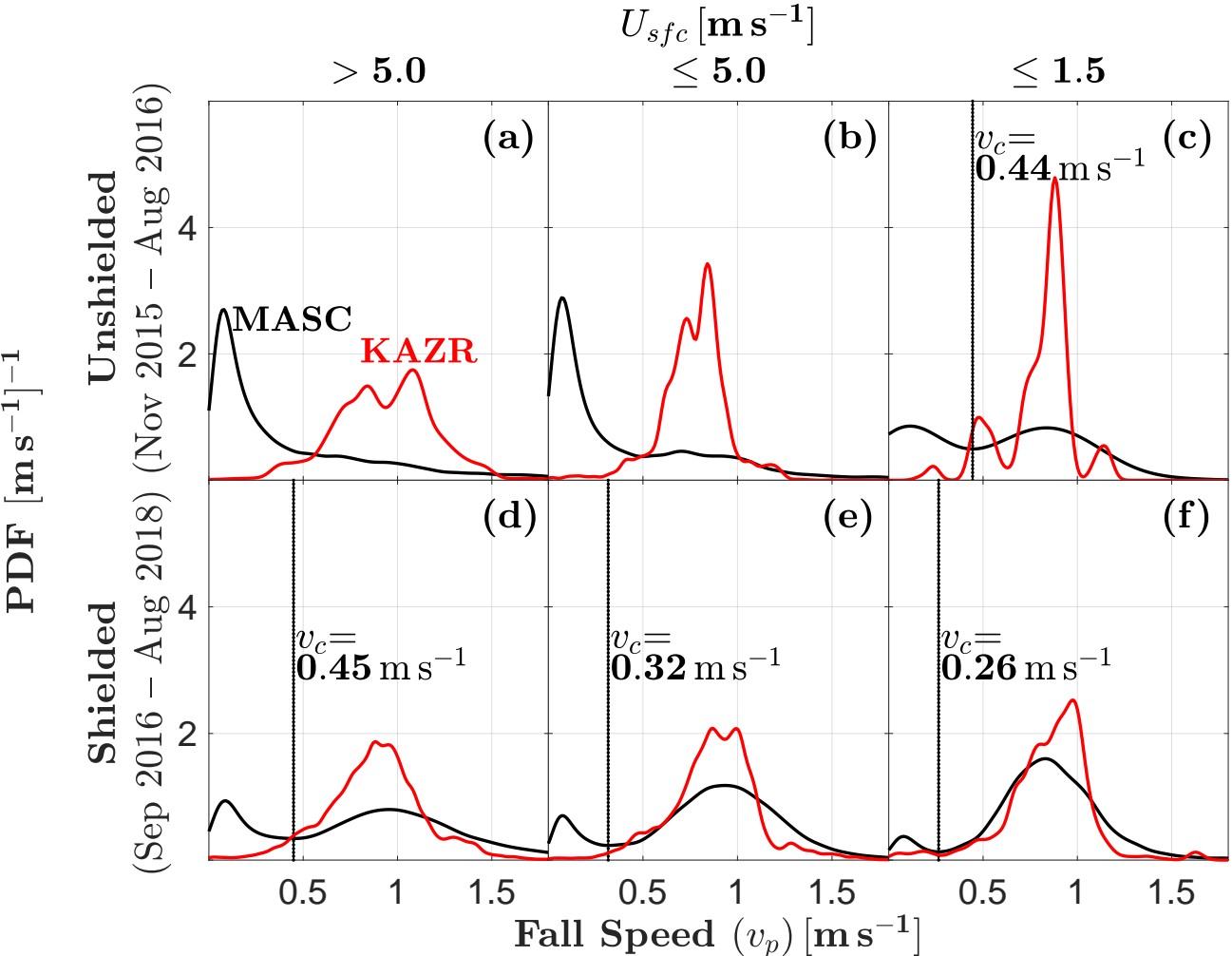

**Figure 5.** Comparison of MASC hydrometeor fall speed and KAZR mean Doppler velocity PDF estimates for (a)–(c) unshielded and (d)–(f) shielded MASC measurements. Surface wind speeds $U_{sfc}$ decrease from left to right. Where MASC PDFs are bimodal, the vertical line marks the cutoff fall speed $v_c$, indicating the location of the local minimum separating the two modes. The number of observations for each case is listed in Table 1. The terms "shielded" and "unshielded" refer only to the MASC. Unshielded and shielded MASC observations are from two separate periods: 29 November 2015 to 21 August 2016 and 22 August 2016 to 28 August 2018, respectively.

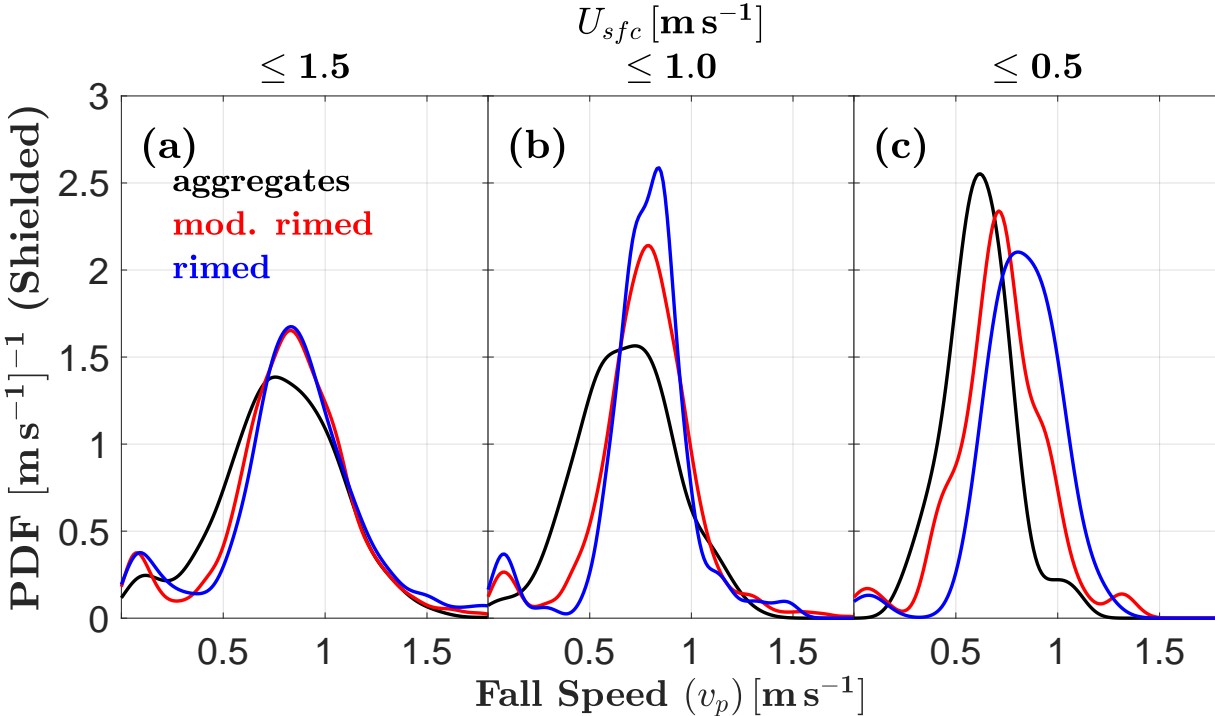

**Figure 6.** PDF estimates for shielded MASC fall speed $v_p$ for very light wind speeds and hydrometeors divided into three riming classes: sparsely-rimed aggregates, moderately rimed, and rimed.

To examine the influence of surface wind speeds on MASC fall speed measurements, Fig. 4(a) shows PDF estimates of wind speed $U_{sfc}$ for the two separate parts of the shielded MASC fall speed distribution from Fig. 3. From the difference (Fig. 4(b)), it is apparent that the high-speed mode of $v_p > 0.45\,\mathrm{m\,s^{-1}}$ is more likely to be observed when $U_{sfc} < 5\,\mathrm{m\,s^{-1}}$.

Figure 5 compares MASC fall speed and KAZR mean Doppler velocity distributions as a function of $U_{sfc}$, again both with and without wind shielding of the MASC. Qualitatively, the agreement between the MASC and KAZR distributions is maximized for shielded MASC measurements with light winds ($U_{sfc} \leq 1.5\,\mathrm{m\,s^{-1}}$), where only 7% of measured fall speeds are lower than the $v_c$ threshold of $0.26\,\mathrm{m\,s^{-1}}$ (Fig. 5(f)). When separated by riming class (Fig. 6), shielded MASC fall speed distributions show discernible differences only for the lightest winds. This is most apparent for $U_{sfc} \leq 0.5\,\mathrm{m\,s^{-1}}$ (Fig. 6(c)), where the most heavily rimed particles ($\chi \leq 1.35$) tend to exhibit the highest fall speeds. Particle counts corresponding to Figs. 5 and 6 are listed in Table 1.

### 2.3 Observations of orientation, maximum dimension, and riming degree

Distributions of unshielded MASC-measured orientation angles tend to favor high angles in high winds ($U_{sfc} > 5\,\mathrm{m\,s^{-1}}$, Fig. 7(a)), where the mode is $57°$, but this shifts to $28°$ for the lightest winds ($U_{sfc} \leq 1.5\,\mathrm{m\,s^{-1}}$, Fig. 7(c)). Shielded measurements

**Table 1.** Number and percentage of observed hydrometeors in each wind shielding case, surface wind speed $U_{sfc}$ category, and riming class. Whole numbers in parentheses indicate the number of distinct events for each category. For each wind-shielded riming category, * indicates where the difference in the complexity ($\chi$) distribution for each low-wind case is statistically significant at the 5% level from that of the respective high wind case ($U_{sfc} > 5\,\mathrm{m\,s^{-1}}$). Percentages may not add to precisely 100% due to rounding. Less restrictive wind speed categories (e.g., $\leq 5\,\mathrm{m\,s^{-1}}$) include data from more restrictive categories (e.g., $\leq 1.5\,\mathrm{m\,s^{-1}}$).

| Category | $> 5\,\mathrm{m\,s^{-1}}$ | $\leq 5\,\mathrm{m\,s^{-1}}$ | $U_{sfc}$ $\leq 1.5\,\mathrm{m\,s^{-1}}$ | $\leq 1.0\,\mathrm{m\,s^{-1}}$ | $\leq 0.5\,\mathrm{m\,s^{-1}}$ |
|---|---|---|---|---|---|
| **No Wind Shield** | **2,249 (27)** | **5,097 (31)** | **460 (9)** | **167 (7)** | **32 (4)** |
| Aggregates | 176 (8%,16) | 1,522 (30%,22) | 67 (15%,6) | 15 (9%,5) | 5 (16%,2) |
| Moderately Rimed | 1,209 (54%,25) | 2,891 (57%,27) | 315 (68%,8) | 115 (69%,6) | 14 (44%,4) |
| Rimed | 864 (38%,13) | 684 (13%,19) | 78 (17%,5) | 37 (22%,2) | 13 (41%,2) |
| **Wind Shield** | **85,151 (181)** | **58,939* (140)** | **5,730* (45)** | **1,372* (30)** | **161* (13)** |
| Aggregates | 15,320 (18%,132) | 11,304 (19%,101) | 1,299 (23%,30) | 302* (22%,21) | 41* (25%,8) |
| Moderately Rimed | 47,147 (55%,165) | 35,820* (61%,128) | 3,477* (61%,38) | 855* (62%,26) | 86 (53%,12) |
| Rimed | 22,684 (27%,151) | 11,815* (20%,107) | 954* (17%,35) | 215* (16%,21) | 34 (21%,6) |

tend towards even lower angles in the lightest winds, with a mode of 12° for $U_{sfc} \leq 1.5\,\mathrm{m\,s^{-1}}$ (Fig. 7(f)). These results suggest that these solid hydrometeors tend to fall with their major axes nearly aligned with the horizontal plane when left undisturbed by surface winds. When separated by riming class for the lightest winds ($U_{sfc} \leq 1.5\,\mathrm{m\,s^{-1}}$), shielded MASC orientation angles tend to be larger for sparsely-rimed aggregates (Fig. 8), meaning their major axes are more often oriented further away from the horizontal plane.

To examine surface wind influence on hydrometeor sizes observed by the MASC, distributions of $D_{max}$ and corresponding $\lambda$ values are shown in Fig. 9. The slope parameter $\lambda$ is smallest when the MASC is shielded and surface winds are very light ($U_{sfc} \leq 1.5\,\mathrm{m\,s^{-1}}$, Fig. 9(f)), and largest for unshielded observations in high winds ($U_{sfc} > 5\,\mathrm{m\,s^{-1}}$, Fig. 9(a)). This suggests that the largest hydrometeors are less likely to be captured by the MASC in strong winds, and even less likely without shielding. When these wind-shielded distributions are separated into riming degree classes (Fig. 10), aggregates exhibit a 26% percent decrease in $\lambda$, from 0.88 to 0.65 mm$^{-1}$, when comparing the case with high winds ($U_{sfc} > 5\,\mathrm{m\,s^{-1}}$) to that with low winds ($U_{sfc} \leq 1.5\,\mathrm{m\,s^{-1}}$). For a size distribution of form $n_{D_{max}} = n_{D_0} \exp(-\lambda D_{max})$, where $n_{D_{max}} \Delta D_{max}$ is the concentration of particles with sizes between $D_{max}$ and $D_{max} + \Delta D_{max}$, this decrease in $\lambda$ corresponds to a number concentration that is 5 times higher for aggregates with $D_{max} = 7\,\mathrm{mm} \pm \Delta D_{max}/2$. In contrast, moderately and heavily rimed hydrometeors only exhibit decreases in $\lambda$ of 13% and 11%, respectively, when comparing high- and low-wind measurements.

The observation that measured concentrations of larger aggregates are relatively sensitive to surface winds compared to more heavily rimed particle types suggests that the frequency distribution of riming classes observed by the MASC might also

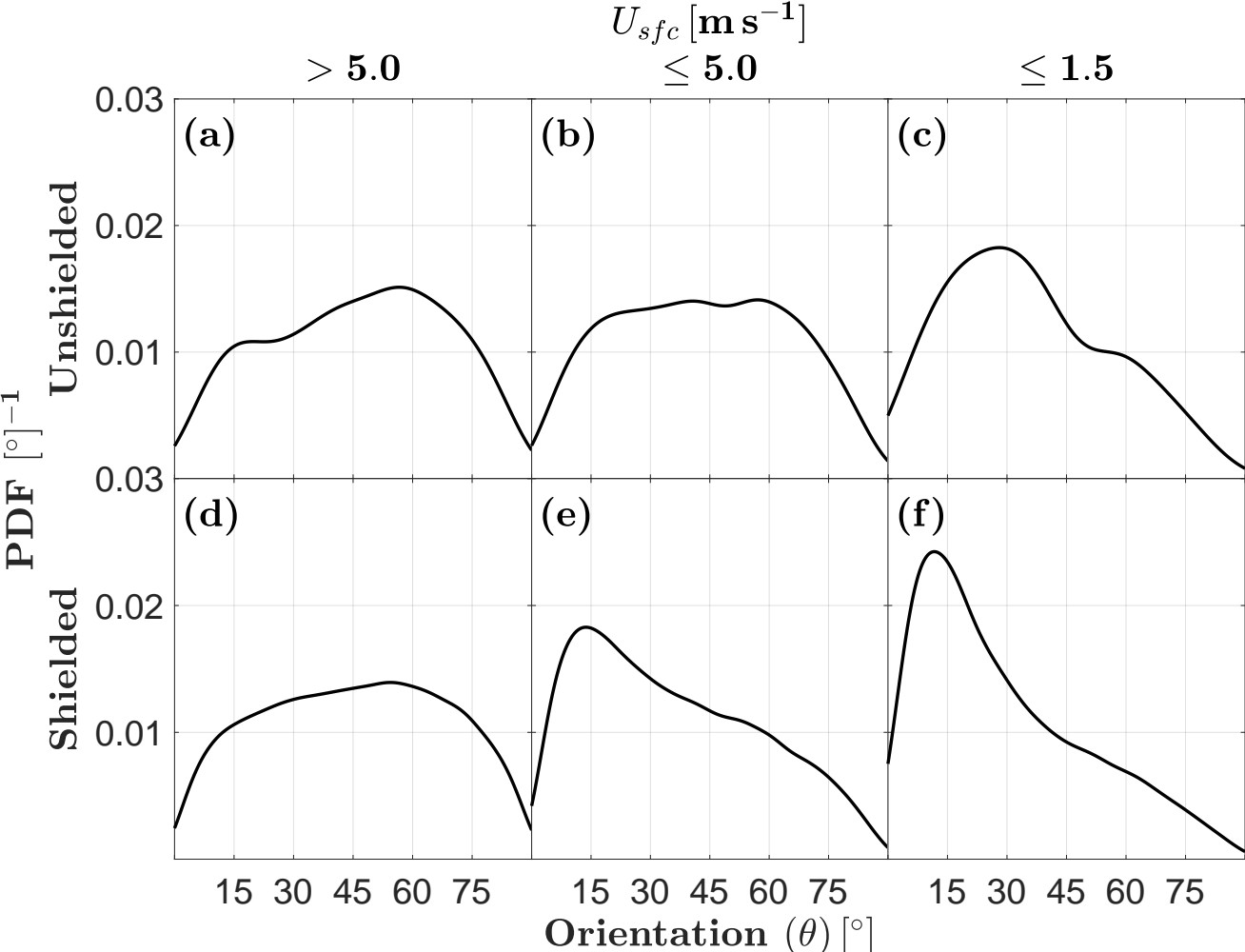

**Figure 7.** Probability distribution function (PDF) estimates of MASC-observed orientation angle $\theta$ as a function of surface wind speed $U_{sfc}$ for both shielded and unshielded configurations. Unshielded and shielded MASC observations are from two separate periods: 29 November 2015 to 21 August 2016 and 22 August 2016 to 28 August 2018, respectively.

reflect this sensitivity. Indeed, Table 1 shows that the percentage of wind-shielded aggregates reaches a maximum (25%) when wind speeds are lowest ($U_{sfc} \leq 0.5\,\mathrm{m\,s^{-1}}$). The opposite is true for shielded rimed hydrometeors (i.e., graupel), implying that high-density rimed particles are more likely to be observed by the MASC than large, weakly rimed aggregates in the presence of strong winds ($U_{sfc} > 5\,\mathrm{m\,s^{-1}}$).

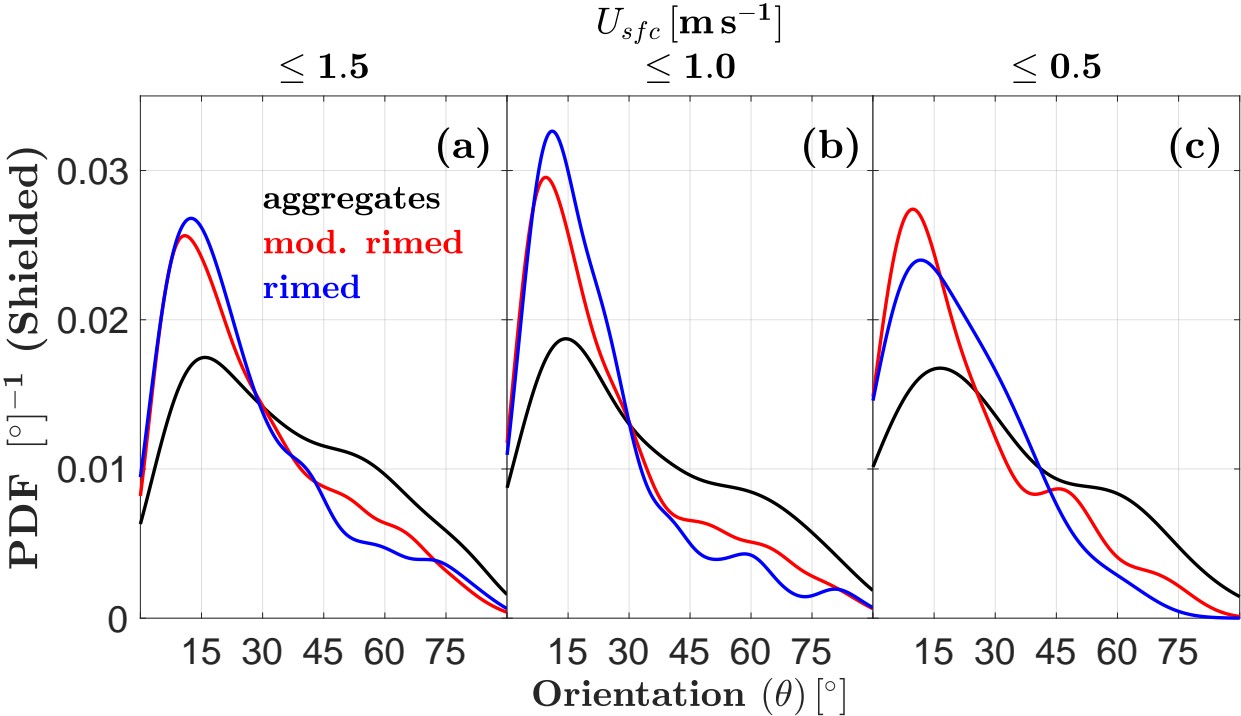

**Figure 8.** As in Fig. 7 but with lighter winds and hydrometeors divided into three riming degree categories: sparsely-rimed aggregates, moderately rimed, and rimed. Only shielded MASC measurements are shown.

## 3 CFD simulations

To explore the fluid-particle-MASC dynamics involved in the influence of ambient winds on MASC measurements of fall speed, we use the OpenFOAM 4.1 tool (Jasak et al., 2007) for CFD calculations of falling particles and winds interacting with the MASC body. OpenFOAM is an open-source CFD toolbox based on C++ libraries and codes designed to solve complex flow dynamics problems (Jasak et al., 2007; Chen et al., 2014; Greenshields, 2015). The incompressible, robust *simpleFoam* solver for steady incompressible turbulent flows (Balogh et al., 2012; Higuera et al., 2014) uses the factorized finite volume method (FVM) with the Semi-Implicit Method for Pressure Linked Equations (SIMPLE) algorithm (Caretto et al., 1973) to solve the Navier–Stokes equations. The $k$–$\omega$ Shear Stress Transport (SST) model is utilized in this study to solve the turbulence closure problem due to its capability to capture the flow separation near objects through the viscous sub-layer, without additional wall functions (Menter, 1993). We combine *simpleFoam* with the *solidParticle* and *solidParticleCloud* classes to study the motions of particles (Iudiciani, 2009). The integrated, semi-developed *solidParticleFoam* is used to simulate particle trajectories, with gravity included to supplement the developed simulation.

To study particle-air interactions, the first step is to determine the two-phase flow type. The ratio between the average inter-particle distance and the particle diameter is estimated. Provided the ratio is $\gtrsim 100$, the flow can be treated as a dilute,

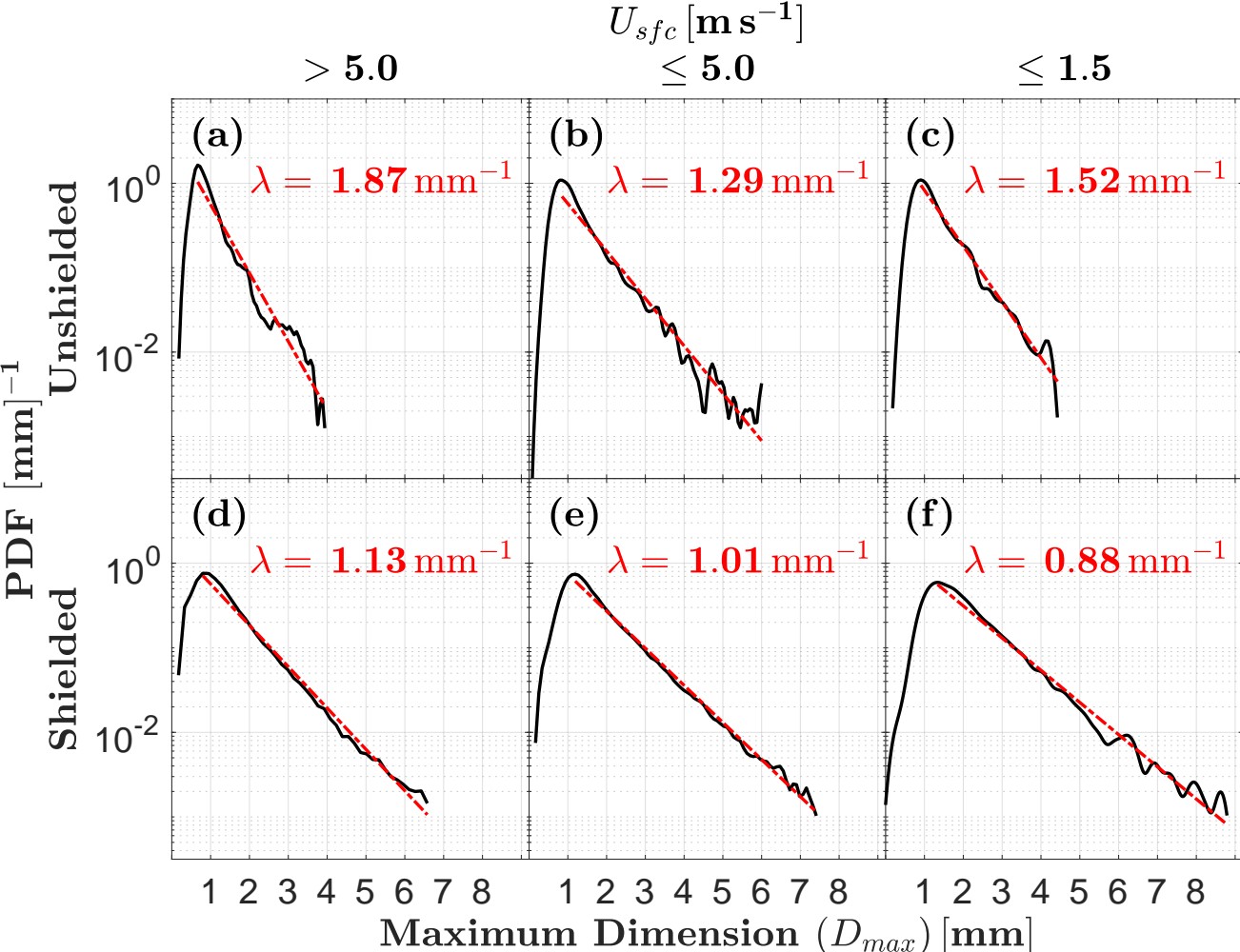

**Figure 9.** As in Fig. 7 but for maximum dimension $D_{max}$ and slope parameter $\lambda$. The slope parameter is calculated as the linear least-squares fit from the peak through the tail of the distribution. Unshielded and shielded MASC observations are from two separate periods: 29 November 2015 to 21 August 2016 and 22 August 2016 to 28 August 2018, respectively.

dispersed system, and one-way coupling — wherein the particles do not collide with each other and also do not affect the flow field — can be assumed (Elghobashi, 1994). The OpenFOAM *blockMesh* and *snappyHexMesh* tools are applied here to generate a mesh around the complex physical geometry of the MASC instrument (Gisen, 2014). The *snappyHexMesh* utility automatically generates 3D meshes containing hexahedra and split-hexahedra from a triangulated surface (MASC in this case).

Figure 11(c–e) shows the MASC mesh for different viewing angles. Spatial and temporal parameters are provided in Table 2.

The *snappyHexMesh* tool requires an existing base mesh to work with, which is generated from *blockMesh* and is represented in Table 2. For *snappyHexMesh*, two of the most important parameters are *nCellsBetweenLevels*, set to 3, and the

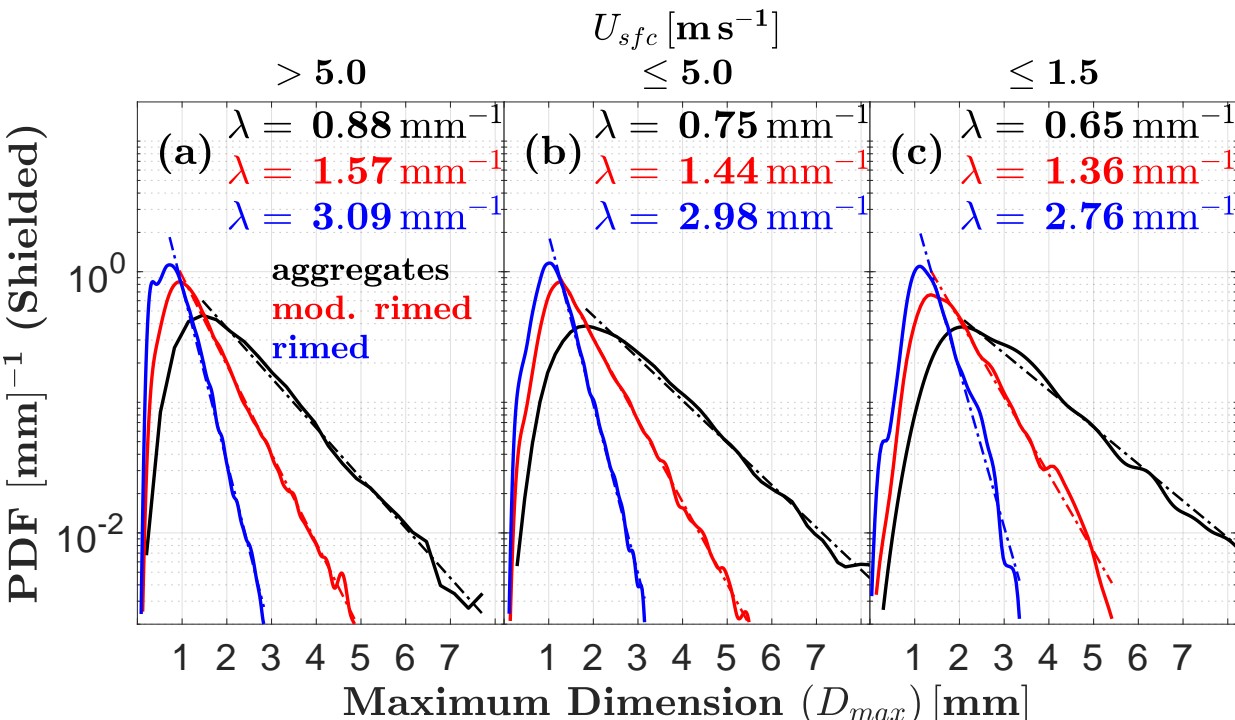

**Figure 10.** As in Fig. 9 but with hydrometeors divided into three riming degree categories: sparsely-rimed aggregates, moderately rimed, and rimed. Only shielded MASC measurements are shown.

*refinementSurfaces* level, which is set to a minimum of 4 and maximum of 5. This brings the total number of cells to 131,864 when the block is 4 m × 4 m × 5 m. These values were determined through analysis of grid independence. For *blockMesh*, the resolution of 25 cm × 25 cm × 25 cm provided the most efficient mesh for a fixed *snappyHexMesh*. The *snappyHexMesh* parameters were also determined through testing; lower values (e.g., *nCellsBetweenLevels* < 3 or *refinementSurfaces* level < 4) rendered the mesh too coarse to capture the interaction between particles and flow inside of the aperture, while larger values come at a much higher computational cost.

For the simulation of hydrometeors in the atmosphere, we track spherical particles of mass $m_p$, diameter $D_p$, and area $A_p$ within a Lagrangian framework, where the Eulerian fluid velocity field $\boldsymbol{v}_f = v_{f_x}\hat{x} + v_{f_y}\hat{y} + v_{f_z}\hat{z}$ is interpolated from nearby grid points at the position of the particle to compute the instantaneous particle drag. The particle velocity $\boldsymbol{v}_p$ is calculated at each time step by assuming that the particle's Reynolds number $Re_p$ is greater than unity, which gives a semi-empirical form of the Maxey–Riley equation of motion (Maxey and Riley, 1983):

$$m_p \frac{d\boldsymbol{v}_p}{dt} = m_p \boldsymbol{g} - \frac{1}{2}\rho_f A_p C_D(Re_p)|\boldsymbol{v}_p(t) - \boldsymbol{v}_f(t)|(\boldsymbol{v}_p(t) - \boldsymbol{v}_f(t)) \qquad (2)$$

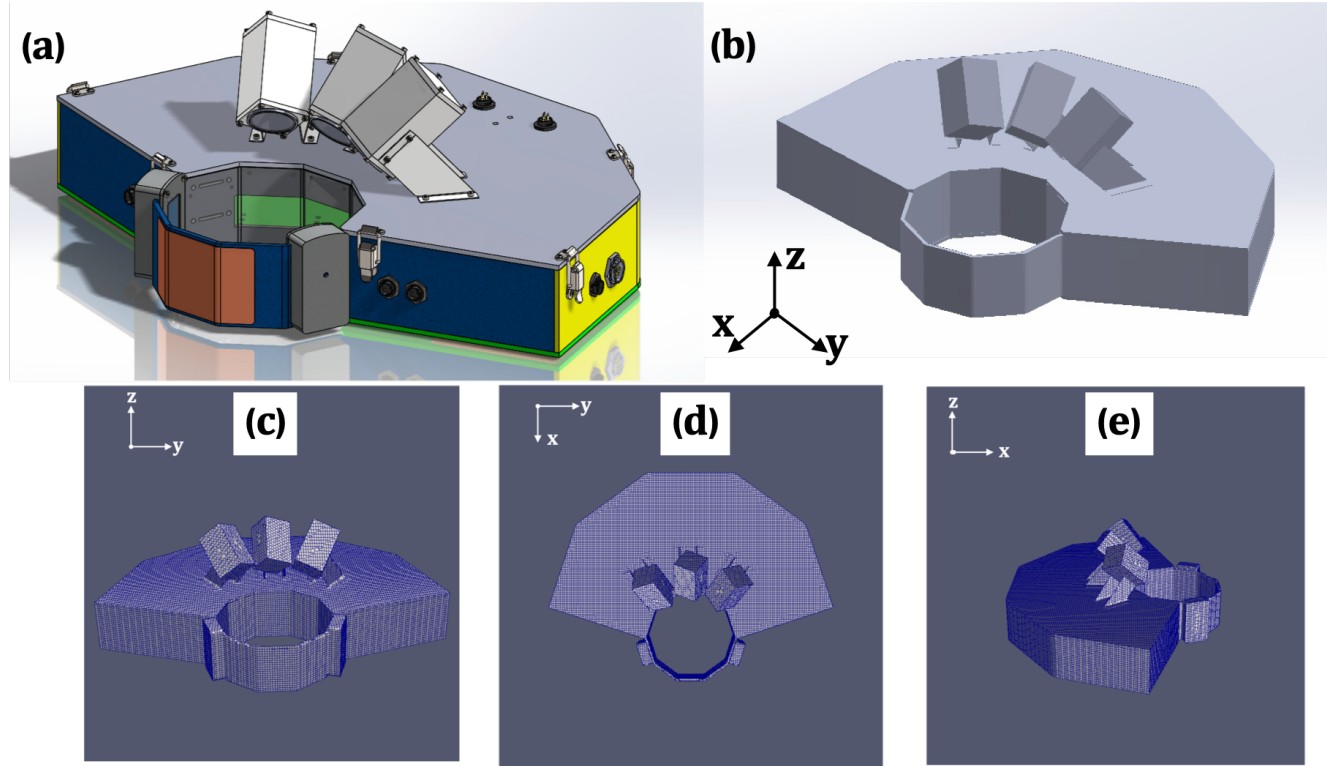

**Figure 11.** (a) original MASC model as a Stereolithography (STL) file; (b) MASC model neglecting small-scale details (e.g., bolts, holes, patches, etc.); (c)–(e) snapped mesh on MASC in three viewing directions.

where the steady drag force $F_d$ changed from scaling linearly with relative velocity to scaling as an empirically derived steady drag coefficient $C_D$ and the relative velocity squared: $F_d = \frac{1}{2}\rho_f A_p C_D(\text{Re}_\text{p})(v_p - v_f)^2$. Here $\rho_f$ is the fluid density and $g$ is the gravitational constant. The drag coefficient $C_D(Re_p)$ is defined as

$$C_D = \begin{cases} \frac{24}{Re_p}\left(1 + \frac{1}{6}Re_p^{2/3}\right) & \text{if } Re_p \leq 1000 \\ 0.44 & \text{if } Re_p > 1000 \end{cases} \tag{3}$$

and is a function of the relative Reynolds number $Re_p = (v_p - v_f)D_p/\mu$, where $\mu$ is the kinematic viscosity. Particles measured
with the MASC had a median $Re_p$ of 108, with $95\%$ of the values in the range of $40 < Re_p < 360$. The boundary conditions for velocity include a flat velocity profile at the inlet; slip conditions at top, bottom, front, back and outlet surfaces; and a no-slip condition at the object (MASC). The "inletOutlet" outlet boundary condition was used, which provides a generic outflow condition. Zero gradient pressure fields are applied at all boundaries. The flow is allowed to reach steady-state prior to tracking particles through a "frozen" flow field.

**Table 2.** Domain size and fluid and particle properties of simulations

| Domain Dimensions | |
|---|---|
| Width (x–dir) | 4 m |
| Transverse thickness (y–dir) | 4 m |
| Height (z–dir) | 10 m |
| Grid (x × y × z) | 16 × 16 × 40 |
| **Particle properties** | |
| Number of particles | 400 |
| Diameter ($D_p$) | 2 mm |
| Density ($\rho_p$) | 50 kg m$^{-3}$ |
| **Fluid properties (at 0 °C)** | |
| Viscosity ($\mu$) | 1.34 ×10$^{-5}$ m$^2$ s$^{-1}$ |
| Density ($\rho_f$) | 1.284 kg m$^{-3}$ |

In simulations of the response of the particles to horizontal winds in the vicinity of the MASC, the particles are evenly distributed on a 20×20 grid with 1 mm spacing in the x–direction and 2 mm spacing in the y–direction. The particles fall downward at an initial velocity of $1\,\mathrm{m\,s^{-1}}$ from an initial height of 3 m above the MASC in the $-z$ direction under the force of gravity, reaching an average terminal velocity of $1.05\,\mathrm{m\,s^{-1}}$ well before encountering flows perturbed by the MASC. Initial particle positions are ~2 to 20 m away from the MASC in the upstream horizontal direction, depending on the flow velocity.
These initial positions were evaluated to ensure the particles fell into the center of the aperture. An example of simulated particle trajectories is shown in Fig. 12.

Figure 13 shows interactions of a horizontal flow in the +y direction of 1 m s$^{-1}$ with the MASC body. There is a clear separation of flow at the upstream side of the aperture, a relatively large upward component above the aperture at the upstream side, and a smaller downward component within the aperture. The fall speeds of particles carried into the aperture by the
prevailing flow are decreased by this upward component of the flow, which increases with increasing wind speeds.

The response of particles to these perturbations for horizontal winds in both the $-x$ and $+y$ directions is shown in Fig. 14. The mean particle fall speed within the MASC aperture decreases from $1.07\,(1.04)\,\mathrm{m\,s^{-1}}$ to $0.30\,(0.26)\,\mathrm{m\,s^{-1}}$ as the ambient wind speed increases from 1 to $10\,\mathrm{m\,s^{-1}}$ (Table 3). Although there is little difference between the wind directions shown in Fig. 14, particles carried by flow in the $+x$ direction were mostly blocked by the LEDs located on top of the MASC, especially
for speeds of $> 2\,\mathrm{m\,s^{-1}}$ (not shown).

The influence of ambient turbulent intensity expressed as $TKE = \frac{1}{2}\left(\overline{v'_{f_x}}^2 + \overline{v'_{f_y}}^2 + \overline{v'_{f_z}}^2\right)$ was calculated for $TKE = 1$, 3, and 5 m$^2$ s$^{-2}$, where the perturbation velocity $v'_f$ is the difference between the instantaneous and average velocities of the atmospheric flow. These TKE values are used as initial conditions in the $k$–$\omega$–$SST$ closure model, which determines the shear

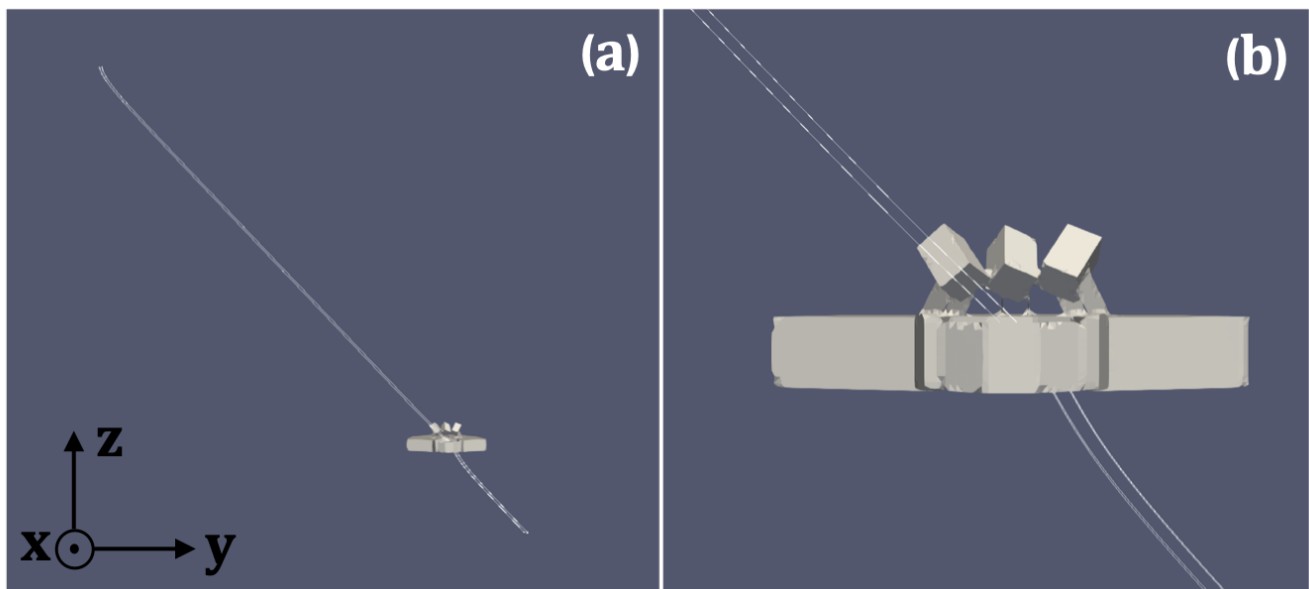

**Figure 12.** Example of simulated particle trajectories for a horizontal wind speed of $1\,\mathrm{m\,s^{-1}}$

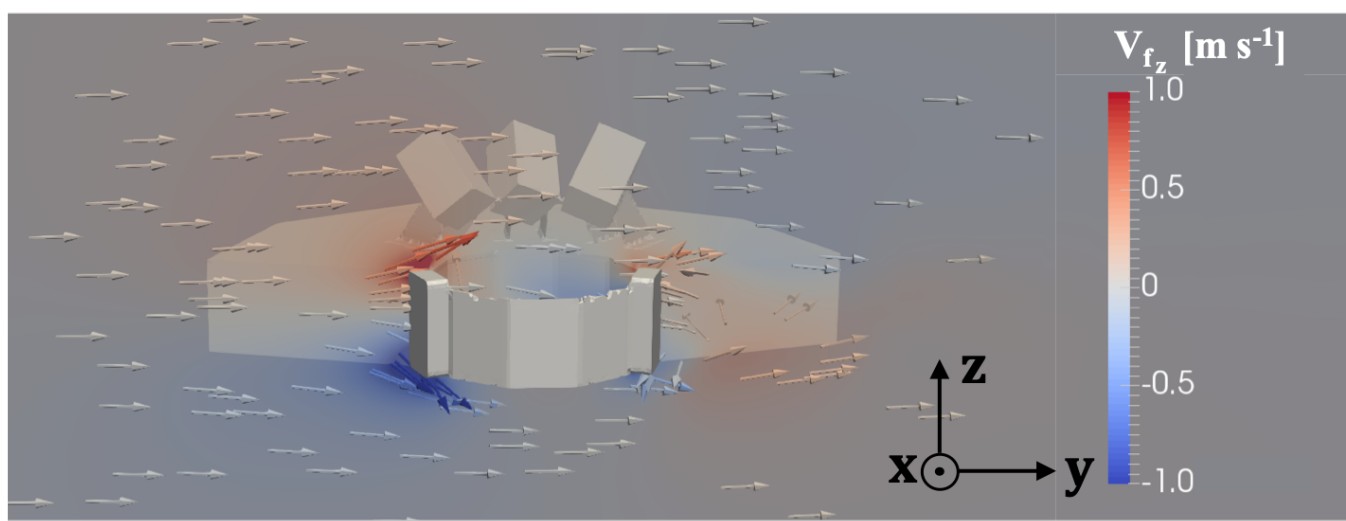

**Figure 13.** Simulated wind field around the MASC with undisturbed winds set at $1\,\mathrm{m\,s^{-1}}$ towards the +y direction. Color represents the vertical wind speed $v_{f_z}$, and arrows show wind directions on the y–z plane. The plane in which the arrows are located is aligned with the center of the aperture on the y–z plane, and x–positive points out of the page.

stress, which in turn is used in the momentum budget equation. Figure 15 shows that for a wind speed of $10\,\mathrm{m\,s^{-1}}$, the mean
particle fall speed is 24% lower for an initial value of $TKE = 1\,\mathrm{m^2\,s^{-2}}$ than it is for $TKE = 5\,\mathrm{m^2\,s^{-2}}$ (Table 3).

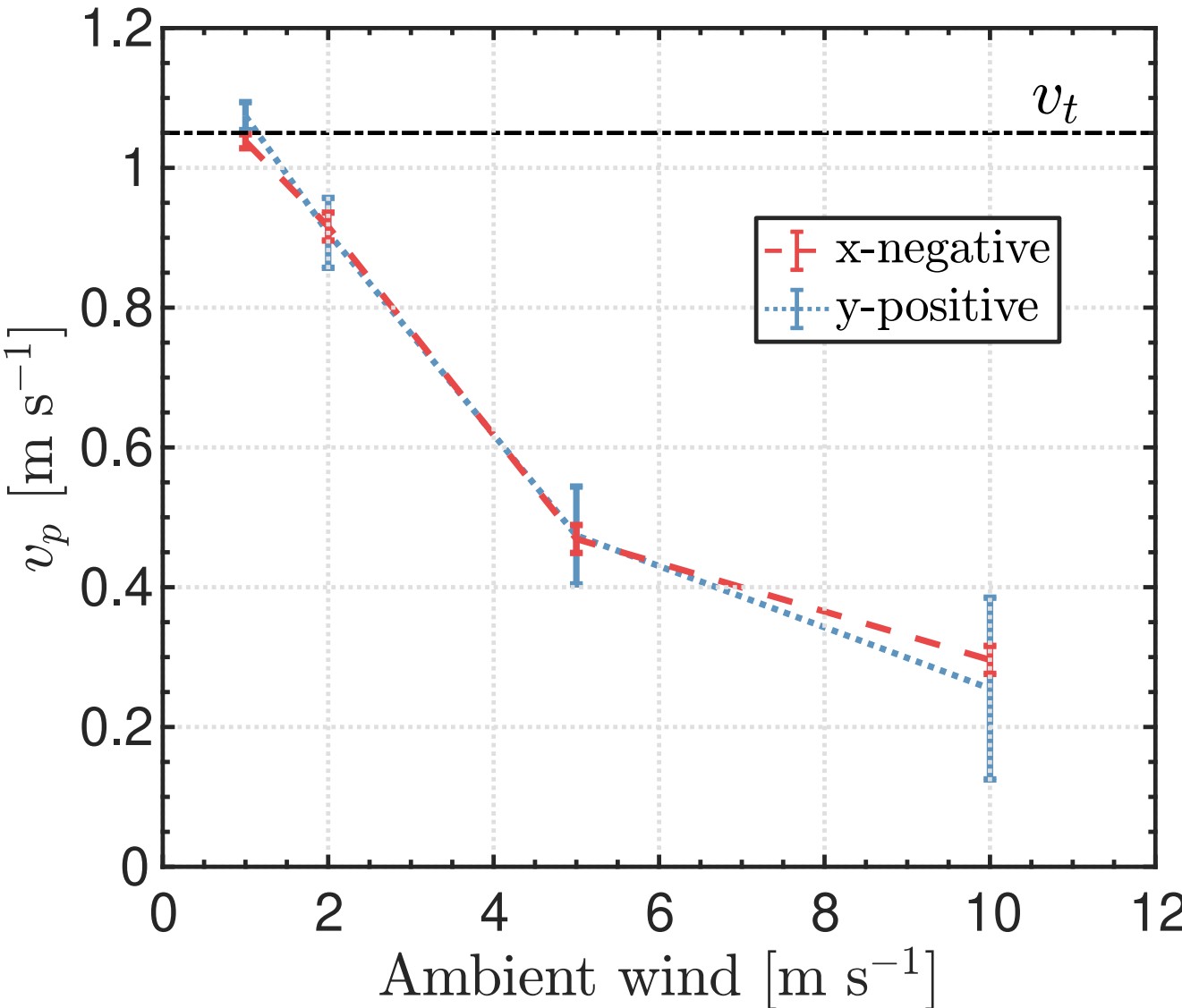

**Figure 14.** Mean fall speed of particles $v_p$ as a function of ambient wind speed. Error bars represent the standard deviation of all the particles at each ambient wind speed. x–negative and y–positive represent the wind pointing towards $-x$ and $+y$ directions (see Fig. 11(d)), respectively. Terminal fall speed $v_t$ is included for comparison, and the initial TKE is $1 \, \mathrm{m^2 \, s^{-2}}$. Data are sampled at the center of the aperture.

## 4  Discussion

The cutoff fall speed $v_c$ defined in Sect. 2.2 is a potentially useful threshold for quality control of MASC fall speed measurements, and Fig. 5 suggests that $v_c = v_c(U_{sfc})$ for shielded MASC measurements. Least squares linear regression fits of $v_c$ to $U_{sfc}$ are plotted in Fig. 16 in increments of $0.5 \, \mathrm{m\,s^{-1}}$. Goodness-of-fit is 0.95 or greater for all but the most heavily rimed

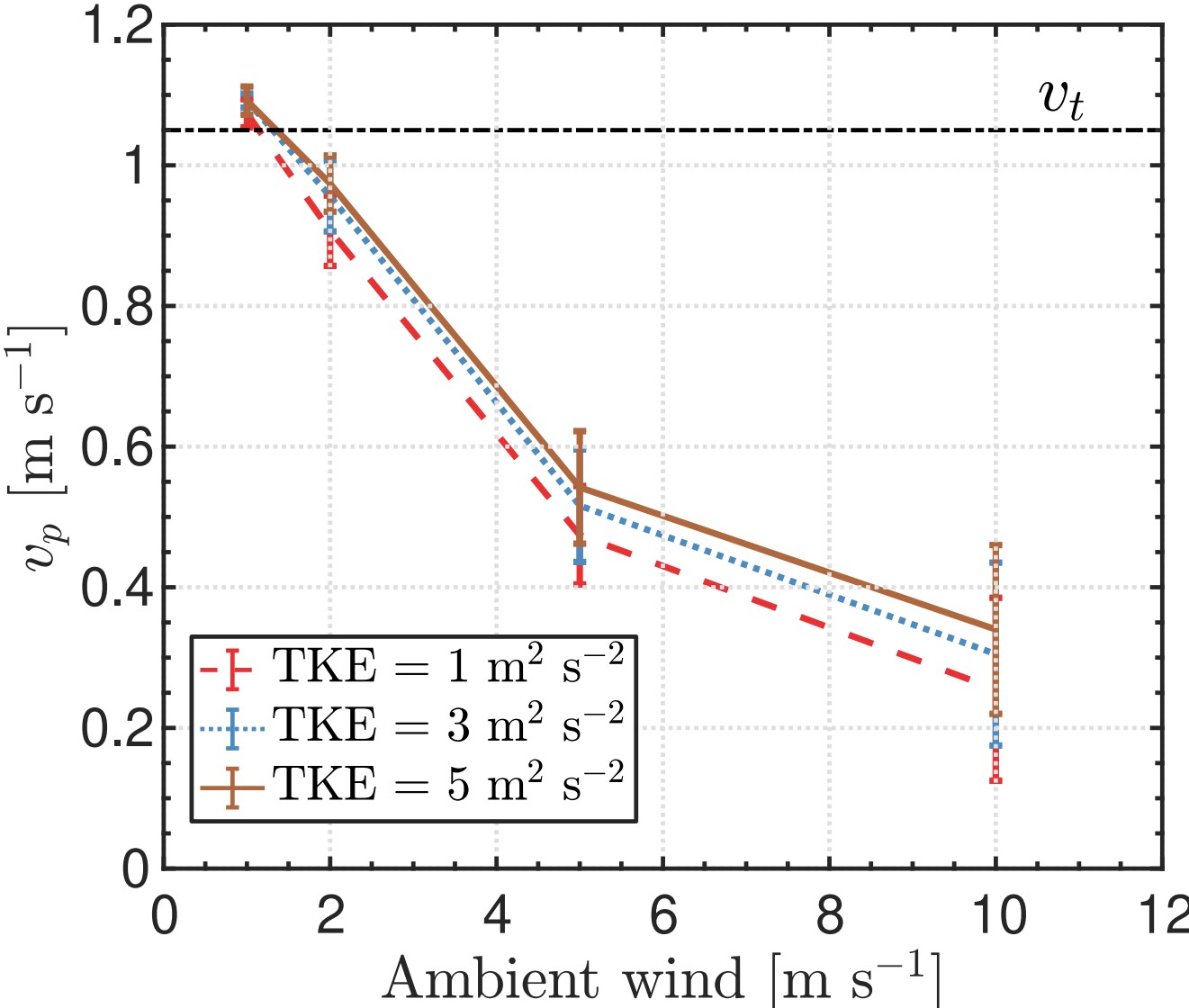

**Figure 15.** Mean fall speed $v_p$ of particles versus ambient wind speed for different values of initial TKE. Terminal fall speed $v_t$ is included for comparison. Data are sampled at the center of the aperture.

particles, where a value of 0 indicates no relationship, and 1 indicates a perfect relationship. Data points tend to fall outside the 95% confidence interval for the most restricted wind speeds ($U_{sfc} < 2\,\mathrm{m\,s^{-1}}$, or $< 4\,\mathrm{m\,s^{-1}}$ for graupel), corresponding to the lowest number of observations. These fits can be used as a guide for quality control of shielded MASC measurements, where particles with fall speeds below $v_c$ are either omitted or corrected through extrapolation.

      For unshielded MASC measurements, the simulations show that the separation of flow leads to an upward flow velocity 265    component above the aperture that tends to decrease the mean fall speed of particles falling into the aperture (Figs. 14 and 15).

**Table 3.** Mean particle fall speed for various wind directions, wind speeds, and TKE values. The terminal fall speed is $1.05\,\mathrm{m\,s^{-1}}$ in all runs.

| Ambient wind | $1\,\mathrm{m\,s^{-1}}$ | $2\,\mathrm{m\,s^{-1}}$ | $5\,\mathrm{m\,s^{-1}}$ | $10\,\mathrm{m\,s^{-1}}$ |
|---|---|---|---|---|
| **Wind Direction** | | | | |
| x-negative | $1.07\,\mathrm{m\,s^{-1}}$ | $0.92\,\mathrm{m\,s^{-1}}$ | $0.47\,\mathrm{m\,s^{-1}}$ | $0.30\,\mathrm{m\,s^{-1}}$ |
| y-positive | $1.04\,\mathrm{m\,s^{-1}}$ | $0.91\,\mathrm{m\,s^{-1}}$ | $0.47\,\mathrm{m\,s^{-1}}$ | $0.26\,\mathrm{m\,s^{-1}}$ |
| **TKE** | | | | |
| $1\,\mathrm{m^2\,s^{-2}}$ | $1.07\,\mathrm{m\,s^{-1}}$ | $0.91\,\mathrm{m\,s^{-1}}$ | $0.47\,\mathrm{m\,s^{-1}}$ | $0.26\,\mathrm{m\,s^{-1}}$ |
| $3\,\mathrm{m^2\,s^{-2}}$ | $1.09\,\mathrm{m\,s^{-1}}$ | $0.96\,\mathrm{m\,s^{-1}}$ | $0.52\,\mathrm{m\,s^{-1}}$ | $0.31\,\mathrm{m\,s^{-1}}$ |
| $5\,\mathrm{m^2\,s^{-2}}$ | $1.09\,\mathrm{m\,s^{-1}}$ | $0.98\,\mathrm{m\,s^{-1}}$ | $0.54\,\mathrm{m\,s^{-1}}$ | $0.34\,\mathrm{m\,s^{-1}}$ |

As wind speed increases, the mean simulated fall speed decreases, and values do not deviate substantially from the mean (Figs. A1 through A4). In contrast, unshielded measurements of fall speed are highly skewed towards low values (Figs. 5(a) and (b)) and the distribution is bimodal for the lightest winds (Fig. 5(c)). Therefore, while the primary effect of perturbed winds slowing particle fall speeds is generally well represented in the simulations, the details appear to be more complicated in reality.

Larger aggregates with negligible riming tend to be more susceptible than smaller, more dense particles to disturbance by surface winds and associated turbulence, with a tendency for more vertical orientations (Fig. 8), slower fall speeds (Fig. 6), and with a lower frequency of occurrence at higher wind speeds (Table 1) than for other riming classes. The Stokes number is defined as the dimensionless ratio of the particle relaxation time to its terminal velocity in still air $v_t/g$, and a characteristic time of isotropic, homogeneous turbulent flow. Snowflakes with low Stokes numbers tend to follow the flow, becoming trapped

in the vortices with the orientation aligning with the local velocity gradient (Voth and Soldati, 2017). The implication is that large, low-density, aggregate-type hydrometeors — with relatively small values of $v_t$ compared to more heavily rimed particles — have low values of the Stokes number and are more likely to follow the motions of any turbulent flow induced by the MASC aperture. This finding is consistent with prior work by Thériault et al. (2012) who showed that for a Geonor gauge inside a single Alter shield, higher-density, faster-falling hydrometeors are collected most efficiently.

The implication is that particle type needs to be considered when accounting for the effect of wind speed on snow measurements. However, the collection efficiencies for all riming classes sampled in the present study are found to be highly sensitive to winds in the absence of a wind shield. This sensitivity is reduced but still apparent for all but perhaps the very lightest winds $U_{sfc} \leq 0.5\,\mathrm{m\,s^{-1}}$, even when located inside of a double wind fence. This is likely the result of upstream turbulence propagating into the collection area as a result of wind interacting with shield deflector fins, as suggested in Colli et al. (2016a, b).

Considering that the MASC observes one hydrometeor at a time, while the KAZR mean Doppler velocity is the mean value from a volume of scattering hydrometeors, it is certainly possible that at least some of the measurements comprising the low-fall-speed mode of the MASC fall speed distributions are a natural result of turbulence and not caused by the interaction of surface winds with the MASC or MASC-shield configuration. However, without more direct fall speed measurements to

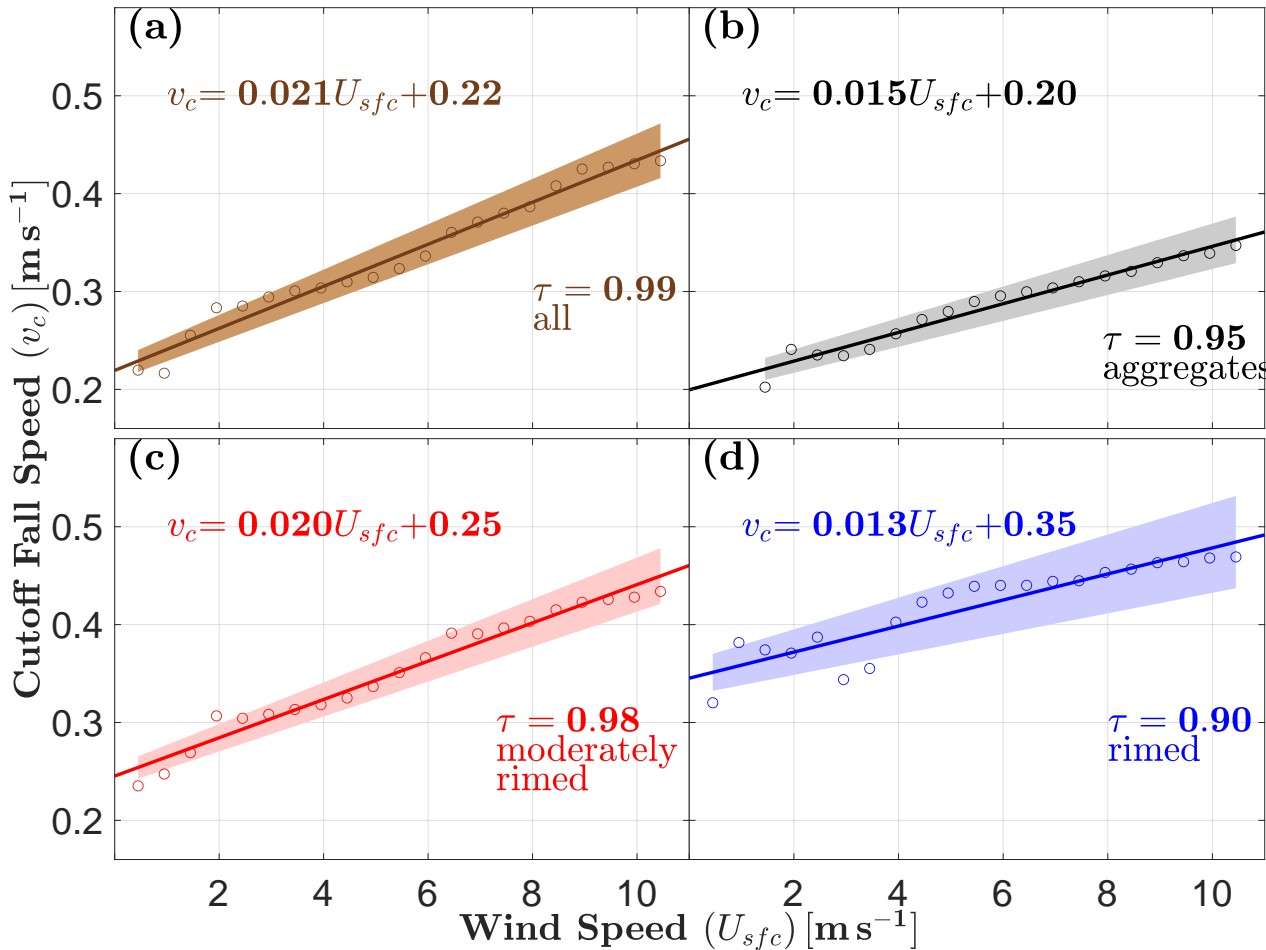

**Figure 16.** Cutoff MASC fall speed $v_c$, defined in Sect. 2.2, as a function of surface wind speed $U_{sfc}$ for (a) all hydrometeor types, (b) aggregates, (c) moderately rimed, and (d) rimed. The solid line in each subplot is a linear least squares best fit, while the shaded regions bound the 95% confidence interval. Goodness-of-fit is measured by applying the Kendall rank correlation coefficient $\tau = 2(P-Q)/n(n-1)$ (Kendall, 1938), where $P$ is the number of concordant pairs, $Q$ is the number of discordant pairs, and $n$ is the total number of pairs. A value of $\tau = 0$ indicates no relationship and 1 indicates a perfect relationship. The confidence interval represents the range of error for predicting a new value for $v_c$. Only shielded MASC measurements are shown.

compare with, the highest confidence in the MASC fall speed measurements is achieved by omitting measured fall speeds that
fall below $v_c$.

For particle values derived from MASC images, the average of all three images was used. An average is not the best guess for the true orientation angle in all cases. For example, depending on the azimuthal orientation with respect to the central camera,

the particle's major axis may not be resolved entirely. Jiang et al. (2019) showed that the azimuthal orientation is correlated with the wind direction, with particles' major axes tending to align with the wind direction. In our case, this would imply that the major axis was often oriented such that it was not entirely resolved by any of the three cameras. More work needs to be done to investigate the limitations of the MASC-determined orientation angle.

## 5  Conclusions

Accurate measurement of solid hydrometeor fall speed, orientation, and size distribution is critical for constraining numerical model parameterizations and remote sensing retrievals. Surface winds are known to have a strong influence on the collection of solid hydrometeors that is dependent on the specific gauge-shield configuration. In comparison with coincident KAZR observations of mean Doppler velocity, MASC measurements of fall speed were in closest agreement only when the MASC was shielded with a double wind fence and winds were light ($U_{sfc} \leq 5\,\mathrm{m\,s^{-1}}$). For the lightest wind speeds ($U_{sfc} \leq 1.5\,\mathrm{m\,s^{-1}}$), shielded measurements of orientation angles decreased to a mode of $12°$, and concentrations of sparsely-rimed aggregates with $D_{max} \simeq 7\,\mathrm{mm}$ increased by a factor of five. However, we showed that even in these wind-restricted and shielded cases, a fraction of MASC-measured fall speeds — those below a wind-speed-dependent cutoff fall speed that is most often $v_c \lesssim 0.5$ — still do not match KAZR measurements. We showed that this cutoff fall speed is a function of wind speed for shielded observations and provided linear regression fits that can be used for additional quality control of MASC measurements.

Simulations of wind interactions with an unshielded MASC yielded an average reduction in mean particle fall speed of 74% for winds increasing to $10\,\mathrm{m\,s^{-1}}$, while TKE had only a weak, inverse effect on the reduction. The simulations revealed that an upward component of perturbed flow at the upstream side of the MASC aperture increases in magnitude with increased wind speed, which in turn leads to a decreased mean particle fall speed.

Relatively simple simulations were carried out here to support the findings of the observations analyses. We used only a single set of particles with limited, yet representative characteristics to support observations analyses with simulated particle responses to MASC-perturbed flow. Future work could include a much more diverse set of particle shapes, sizes, and densities, as well as a turbulent dispersion model and other forces that have been neglected in this work. Furthermore, a double wind fence should be included in future CFD simulations of flow in and around the MASC to see more precisely how the wind field evolves as it encounters the individual deflector fins in each portion of the fence. Thériault et al. (2012) simulated the wind field for a Geonor gauge with a single Alter shield by accounting for the movement of deflector fins on the upstream side of the gauge, where fins were assigned angles with respect to the vertical that increased as a function of wind speed. Such careful simulation might improve the fidelity of wind-shield-gauge influence on snow measurements.

The intent of this work is to provide guidance for under what measurement conditions the MASC can be used to obtain accurate information about hydrometeor microphysical properties and fall speeds. However, those conditions are limited to measurements within still air. The distributions of solid hydrometeor size, type, orientation, and fall speed in natural, turbulent air remain to be determined.

 **Appendix A:  Simulated particle fall speed distributions**

Figures A1, A2, A3, and A4 show simulated particle fall speed distributions for horizontal wind speeds of $1, 2, 5,$ and $10\,\mathrm{m\,s^{-1}}$, respectively.

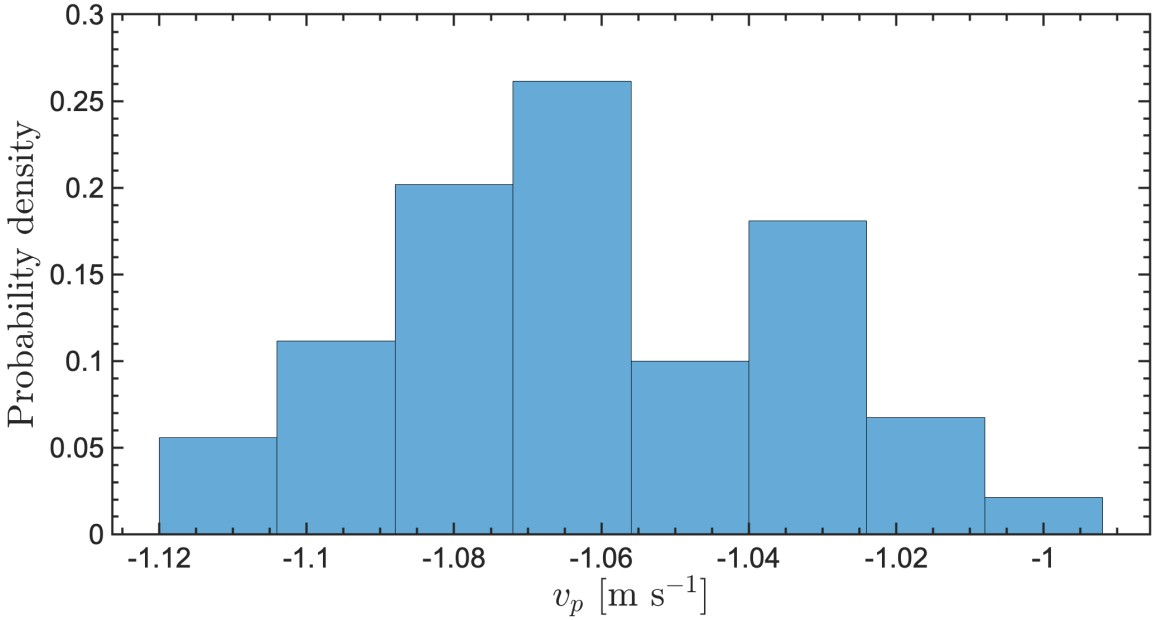

**Figure A1.** Simulated particle fall speed distributions for a horizontal wind speed of $1\,\mathrm{m\,s^{-1}}$

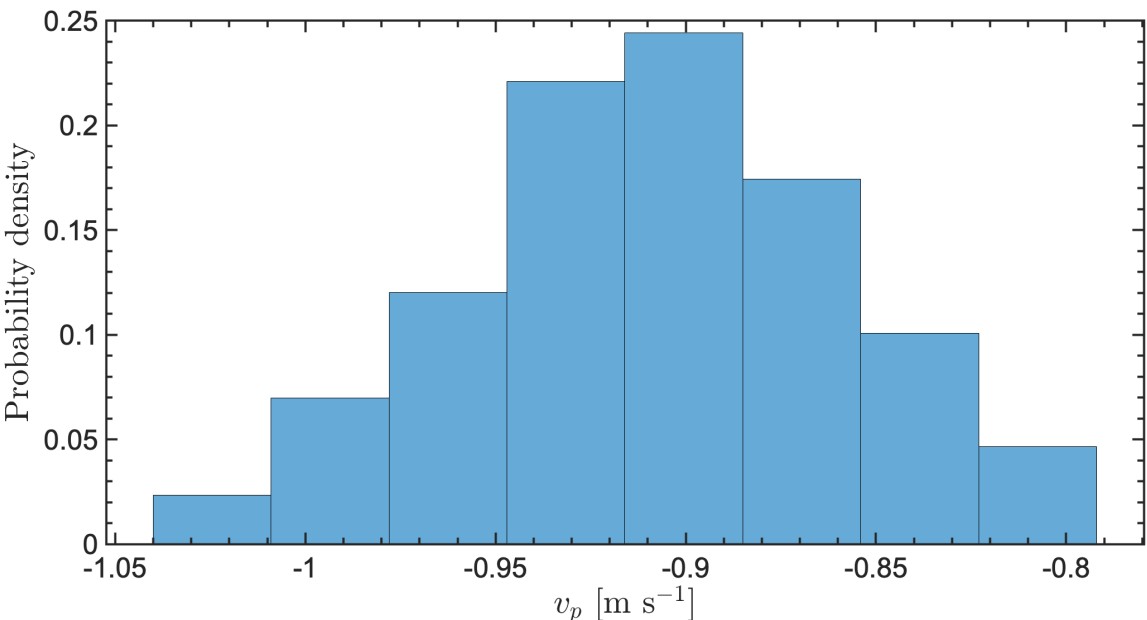

**Figure A2.** Simulated particle fall speed distributions for a horizontal wind speed of $2\,\mathrm{m\,s}^{-1}$

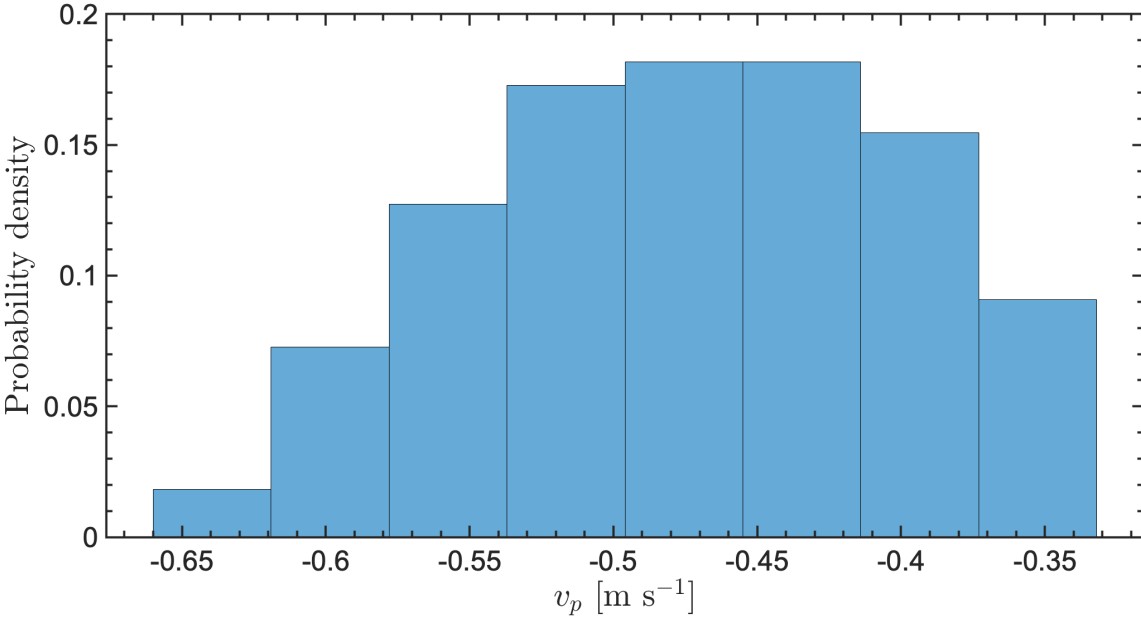

**Figure A3.** Simulated particle fall speed distributions for a horizontal wind speed of $5\,\mathrm{m\,s}^{-1}$

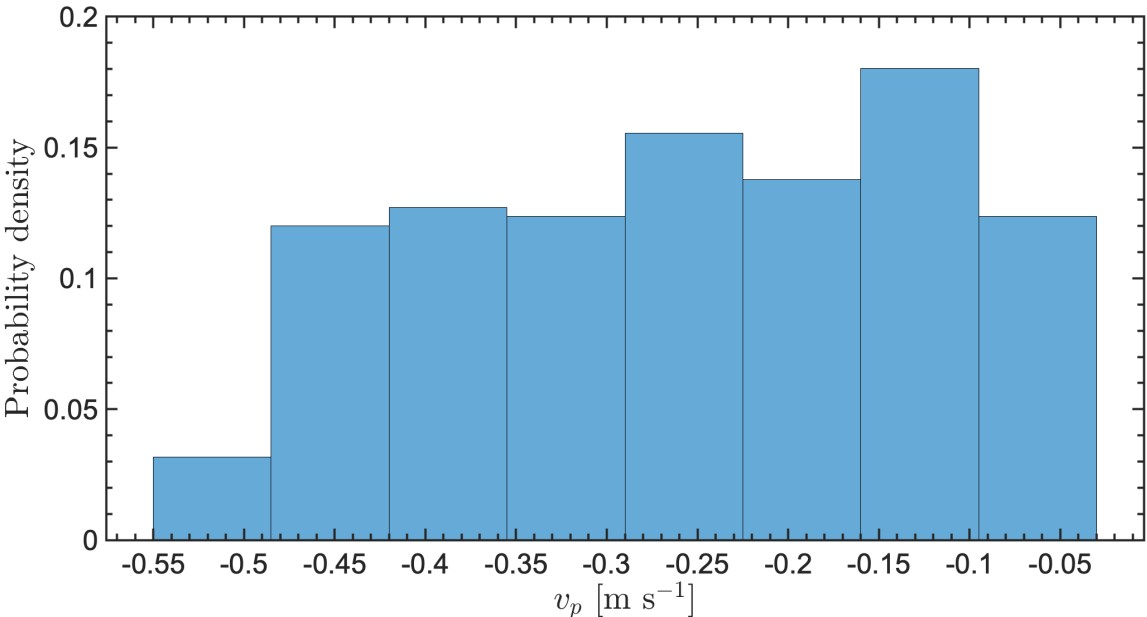

**Figure A4.** Simulated particle fall speed distributions for a horizontal wind speed of $10\,\mathrm{m\,s^{-1}}$

*Code and data availability.* The code and data supporting this project are available at https://doi.org/10.7278/S50DQTX9K7QY. This repository includes code sufficient to replicate the observations analysis results. Raw and processed MASC data are available from the ARM data
archive at https://adc.arm.gov/discovery/#/, and raw MASC data can be processed with the *mascpy* code located at
https://doi.org/10.7278/S50DVA5JK2PD. OpenFoam v4.1 software is available at https://openfoam.org/version/4-1/.

*Author contributions.* All authors contributed to the formulation of the project. KF and TG developed the methodology and software code for observations analysis. CH developed the methodology and software implementation for the simulations, with AT advising. KF and CH wrote the article with contributions from TG and AT.

*Competing interests.* TG is co-owner and scientific advisor of Particle Flux Analytics, Inc., the company manufacturing the Multi-Angle Snowflake Camera. Otherwise, the authors declare that they have no conflict of interest.

*Acknowledgements.* This work was supported by the Department of Energy Atmospheric System Research program Grant DE-SC0016282 and the National Science Foundation (NSF) Physical and Dynamic Meteorology program award number 1841870. We thank Krista Gaustad

and Martin Stuefer for sharing dates, locations, and deployment details of the instrument and wind shield. We also thank Silvio Schmalfuß and
two anonymous reviewers for their comments and questions, which have improved this work substantially.

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
