# Peer review of "Arctic observations and numerical simulations of surface wind effects on Multi-Angle Snowflake Camera measurements"

_Atmospheric Measurement Techniques, 2020_

## Referee Comment (RC1) · Silvio Schmalfuß (Referee) · 24 Sep 2020

The subject of the well written and structured paper is well within the scope of AMT as it provides new tools and data for the correct use of an (already established) instrument for snowflake measurements called "Multi-Angle Snowflake Camera" (MASC). For this purpose, the authors use a combination of numerical simulations based on an Euler-Lagrange approach and field based measurements. The main findings are guidelines for the correct usage of the above mentioned MASC, which I consider a substantial contribution to further investigations of snowfall measurements.

Event though the article is quite good in my opinion and I would not argue with the main

findings, I have some remarks. (Disclaimer: Please note that I am not an expert in precipitation measurement and my expertise mainly is in fluid and particle dynamics and the simulation thereof. So if there are points regarding especially the meteorological topics that I did not get right, please do not hesitate to object/improve.)

Major remarks:

1) The description of the CFD simulation setup should be a bit more detailed: What is the total number of cells? (Table 1 seems to give a wrong information about this, as the cells would be much larger as they seem in Figures 1(c)-1(e)) Have there been any examinations on grid independency? Please specify the boundary conditions. Please justify why potentially important forces acting on particles in turbulent/shear flows are neglected (lift force, pressure gradient force) Where are the starting positions of the particles? (could be included e.g. in one of the figures 1(c)-1(e))

2) Furthermore, I would suggest using more particles and particles of different size and shape according to the different riming classes seen in the measurements. And, as the influence of turbulent wind on the collection characteristics is examined, a model for turbulent dispersion should also be included. The finding that fall velocity is reduced with increasing wind speed could maybe have been made without the particle simulation part by just looking at the fluid velocities inside/above the collection area. If so, the particle simulations might as well be left out and more detailed particle simulations would be something for further work.

3) As in principle the dependence of the collection efficiency on particle dynamic properties is investigated, I would suggest to include some thoughts on and references to general particle dynamics. Namely, there is the particle Stokes number, defining the ability of particles to respond to sudden changes in the carrier fluid's velocity. It is proportional to the product of particle density and the square of the diameter. Aggregates with comparatively low densities would have a low Stokes number, indicating that they follow the flow better than more dense aggregates. Additionally, bodies moving through fluids tend to orient in a manner that the drag is maximised. For oblate particles this would mean that their "plane" is oriented perpendicular to the flow direction. With these two principles a lot of the paper's findings can be explained quite well.

4) Regarding the orientation of particles: As non-spherical particles tend to arrange themselves in some preferential angle to the flow, the concept of determining their orientation seems questionable even for medium wind speeds between 1.5 m/s and 5.0 m/s, as the flow above and inside the measurement region seems to be heavily influenced by the device. What are the authors' views on this?

Minor remarks:

Section 1:

L46f: suggested change from "characterize" to "were used to characterize" or similar

L58f: Isn't the collection region cylindrical or circular, not ring-shaped?

L58ff: I would suggest to include some information about the different parts' positions in Fig. 1a)

L70ff: Maybe the number of references can be reduced, where possible?

L90f: also include influence on riming degree among "fall speed, fall orientation, and size distribution"?

Section 2:

Please specify more clearly: It seems that first a steady state simulation with simple-Foam is done, and afterwards particles are tracked through a "frozen" flow field, but it is not actually mentioned.

L96: Which solver? Maybe simpleFoam should be mentioned here already.

L100: simpleFoam instead of simpleFOAM

L115: Please specify the drag coefficient function.

СЗ

**L115 & L117: Re should be Re\_p**

Fig. 2: I would suggest writing "wind directions" instead of "wind vectors". What crosssection is meant? Maybe the plane for the arrows? Please specify. A more detailed view of the results inside/around the measurement section would be nice, as it would be possible to examine the wind speed there more detailed. Is it possible to show some exemplary particle path lines? (additional images could be included as supplementary material)

Fig. 3 and Fig. 4: Please indicate where the data is sampled.

L130: "in the k- $\omega$ -SST closure model"

Section 3:

L137: "the angle between D\_max and the local horizontal" should be "the angle between the major axis and the local horizontal"

L143: Is the average of the three images the best guess for the actual orientation angle? How "stable" is the detection of orientation angles for rimed particles, as they might have no clear major axis?

Fig. 6(b) might be dropped in my opinion

L171: Why were the PDFs adjusted with a Gaussian kernel and how do the original data look?

Table 3: In the regime of U\_sfc <= 5 m/s: Are the data for smaller U\_sfc also contained in values for larger U\_sfc? (e.g. is the data of U\_sfc <= 0.5 m/s also in U\_sfc <= 1.0 m/s?)

L187f: If something different is simulated, then it should not be compared to that. There seems to be a significant difference between the measurements and the simulations. Please include some discussion about this difference.

Fig. 10: The low fall speed mode seems to be less prominent (U\_sfc <= 1.0 and 1.5 m/s) or even vanishes (U\_sfc <= 0.5 m/s) for aggregates. Is there any explanation for this?

Section 4:

As metioned above, please include some discussion of the simulation results and how they compare to the measurements.

As mentioned in major remarks, I suggest including some of these thoughts:

Falling objects tend to orient in maximum drag conditions, which matches the findings here. (compare L196ff)

As aggregates seem to have smaller fall velocities than particles of the other two categories, they should be oriented more vertical to orient in maximum drag conditions when experiencing the same horizontal wind speeds. This again matches with the findings. (compare L201)

Aggregates are more prone to get blown away by the surface winds as they have comparatively low Stokes numbers and high surface-to-density-ratios in comparison to more rimed particles. (compare L205ff)

L236: "enhanced are reduced" should read "enhanced or reduced"

---

## Referee Comment (RC2) · Anonymous Referee #2 · 13 Oct 2020

The manuscript investigates the aerodynamic impact of the particles' fall speed measured by a Multi-Angle Snowflake Camera (MASC) using field measurements and Computational Fluid Dynamics (CFD) simulations. It compares the fall speed PDF measured by the MASC and the K-band radar located at the same site. The distribution of fall speed differs from the two instruments and the numerical simulations suggested that the fall speed measured in strong winds (> 5 m/s) would record slower falling particles when not shielded. Similar results were found using the simulations. Overall, this study helps to improve the quality control procedure of the MASC data and contributes significantly to the field of snowfall measurement. It fits well in this journal as it improves the methodology to be used to quality control MASC data. The

manuscript is very well written and clear. The figures are also clear and well described in the caption. I have, however, a few main and minor comments should be considered before publication.

Main comments

1. The manuscript gives the impression that the main point is the CFD simulations where it would have been used to study the collection efficiency or to develop transfer function to adjust the MASC measurements. After reading the manuscript, the CFD simulations are used only to explain the field measurements. Given that, I think that the authors should add a methodology section after the introduction that explains the approach taken in this study, which includes the field measurements and the simulations. It may also be useful to present the measurements before showing the results from the simulations.

2. More details should be given about the simulations conducted such as, for example, the number and shape of the mesh used. Did you use the integrated trajectory simulations or developed one? Could you add a figure that includes examples of particles' trajectories? In Table 2, only one size of particle was used. For dry snow and aggregates, the fall speed does not change much with diameter according to Rasmussen et al. (1999). However, the fall speed of rimed particles can vary a lot with sizes. Why not use more particles' sizes? What would be the impact on your results? Why did you choose a diameter of 2 mm and not 1 mm? Please also describe in more details the simulations. For example, is there an updraft as found in previous studies (ex: Colli et al. 2016a,b; Theriault et al. 2012) for the Geonor (shielded and not)? How do you explain that slower falling snowflakes fall is detected by the MASC in stronger winds? Add any other details that could help better understanding the results from the simulations.

3. The simulation as well as the measurement shows that an unshielded MASC leads to a decrease of the fall speed. Can you add a brief explanation in the manuscript? It seems counterintuitive as faster falling particles would tend to fall in the gauge in

stronger winds.

4. At lower wind speed, larger aggregates tend to be more detected by the MASC than at higher wind speeds. In theory larger ones would fall faster and would not be deflected. How do you explain this finding? Could it be because larger aggregates in strong winds would breakup? Or is it common to report large aggregates in windy conditions at that site ? Did you compare with the climatology of solid precipitation at that location?

Some minor comments:

1. Lines 5-7: This sentence mentions that the simulations are compared with observations. However, I understood that in this study that the catch efficiency of the instruments is not computed from the simulations and compared with the measured one. But the simulations are used to explain the decrease in fall speed measured in strong winds. This is related to major comment #1. It should be rephrased for clarity.

2. Lines 45: Newman et al. (2009) also conducted CFD simulations in the vicinity of a snowflake video imager. Should probably add the paper to this paragraph. Newman, A. J., P. A. Kucera, and L. F. Bliven, 2009: Presenting the Snowflake Video Imager (SVI). J. Atmos. Oceanic Technol., 26, 167–179, https://doi.org/10.1175/2008JTECHA1148.1.

3. Figures 7, 9, 11 and 13: Those figures compare data taken with an unshielded and a shielded MASC. Please clarify in the caption that the shielded and unshielded data were collected during two different periods.

4. Lines 185: Usfc is defined. Could you explain it further and how it compares with standard measurements of wind speed at the instrument's height (or at 10 m)? I may have missed the explanation in the text.

5. Lines 189-194: The authors forgot to introduce figure 10. Only Figure 10c is referred to.

6. Lines 200-201: Please clarify that sentence. I don't understand what you mean by

'more vertical'?

7. Lines 224-229: For clarity, the authors could remind the reader that larger aggregates fall slower than the rimed particles as in Figure 10.

---

## Referee Comment (RC3) · Anonymous Referee #3 · 15 Oct 2020

**Review of AMT-2020-296**

This manuscript summarizes work focused on snow particle observations from a MASC deployed at the NSA site. Specifically, the authors investigate the impact of wind shielding on the MASC instrumentation both from field measurements and from Computational Fluid Dynamics (CFD) simulations. The authors find that the fall speeds from the MASC agreed much better with Doppler velocities from a co-located KAZR for the wind shielded events. Additionally, the CFD simulations indicated slower particle fall speeds when the MASC was unshielded. In general, I think this work is novel and advances the field of in situ snow particle observations. I would also like to commend the authors on an exceptionally well-written manuscript, which had a clear narrative and was enjoyable to read.

I have one major comment and a few minor that should be addressed prior to publication:

**Major Comment:**

It is unclear to me how many distinct events were used to comprise the observations that were presented. In the methods, the timelines of the MASC deployment unshielded (Feb 2015 - Aug 2016) and shielded (Aug 2016 - Aug 2018) are outlined (33 months total), however it is not discussed anywhere how many independent events are used in this work. This is key information that is missing from this manuscript as it lends weight to the differences seen between unshielded and shielded observations. This is especially true for Table 3 - as the observations are further divided into wind speed bins and by particle type. A single event could produce 1000s of particle images, so it should be made clear how many independent events were used. This should be added to the methods section – ideally as a table (dates, times). Currently, the manuscript reads as if there are enough observations to say that these fractions of different particle types (in Table 3) are due primarily to the wind shielding impacts, however if there is a low number of independent events (or a low number in a represented wind speed range), then some of these particle type ratios could be from different synoptic or thermodynamic forcing. In addition to including the number of events, the authors should also examine the statistical significance of these differences in particle type (rimed, MR, agg.) for the various wind speed bins (if the N of individual events is large enough).

If only a few independent events were used in this work, I think this should be made clear and the language should reflect that is the case. The implication in the paper (whether purposeful or unconscious) is that the differences in particle type ratios seen in shielded versus unshielded at various wind speeds are a product purely from mitigating the wind to the MASC. However, if very few independent snow events were used in these comparisons the synoptic and thermodynamic conditions could be influencing the ratios of rimed, MR, and aggregate particles.

Minor Comments:

Figure 2 illustrates a CFD simulation across the MASC in the +y direction, which is roughly parallel to the cameras that protrude above the opening. And Fig. 3 shows the impact of the fall speeds for ambient winds in both +y and -x (which I read to be winds toward the cameras). What is the impact of the winds originating from behind the cameras, as this is a large obstacle adjacent to the observing ring? I assume that this direction (+x) will have a larger impact on the particle fall speeds (if I am reading the orientation of the axes correctly). Did you do simulations with the wind originating from behind the cameras?

Along those same lines, wind direction impacts were noted in the discussion about the simulations (minimal), but not in the observations. Was there any analysis on the impacts of wind direction from the observational perspective?

The MASC fall speeds were compared to the KAZR Doppler velocities, and it seems that the mean Doppler velocity from the cloud base to near-surface (I assume) profile was used – is that correct? If so, what was the lowest near-surface bin used in the Doppler velocity profile mean calculation (assuming near-surface to cloud base mean value)? Also, the snow particles can change between the cloud base and the surface – so what is the advantage of using the mean DV value of the profile (near-surface to CB) versus simply using the near-surface Doppler velocity? My instinct is that using the near-surface Doppler velocity value would give you a more direct comparison to the MASC

СЗ

---

## Author Comment (AC1) · 27 Nov 2020

The subject of the well written and structured paper is well within the scope of AMT as it provides new tools and data for the correct use of an (already established) instrument for snowflake measurements called "Multi-Angle Snowflake Camera" (MASC). For this purpose, the authors use a combination of numerical simulations based on an Euler-Lagrange approach and field based measurements. The main findings are guidelines for the correct usage of the above mentioned MASC, which I consider a substantial contribution to further investigations of snowfall measurements. Event though the article is quite good in my

[Figure]

**opinion and I would not argue with the main findings, I have some remarks. (Disclaimer: Please note that I am not an expert in precipitation measurement and my expertise mainly is in fluid and particle dynamics and the simulation thereof. So if there are points regarding especially the meteorological topics that I did not get right, please do not hesitate to object/improve.)**

**Major remarks:**

**1) The description of the CFD simulation setup should be a bit more detailed: What is the total number of cells? (Table 1 seems to give a wrong information about this, as the cells would be much larger as they seem in Figures 1(c)-1(e)) Have there been any examinations on grid independency?**

We have added more details to the description as follows (note Table 2 corresponds to Table 1 in the original manuscript, while Figure 11 corresponds to Figure 1):

"The *snappyHexMesh* tool requires an existing base mesh to work with, which is generated from *blockMesh* and is represented in Table 2. For *snappyHexMesh*, two of the most important parameters are *nCellsBetweenLevels*, set to 3, and the *refinementSurfaces* level, which is set to a minimum of 4 and maximum of 5. This brings the total number of cells to 131,864 when the block is 4 m $\times$ 4 m $\times$ 5 m. These values were determined through analysis of grid independence. For *blockMesh*, the resolution of 25 cm $\times$ 25 cm $\times$ 25 cm provided the most efficient mesh for a fixed *snappyHexMesh*. The *snappyHexMesh* parameters were also determined through testing; lower values (e.g., *nCellsBetweenLevels* $< 3$ or *refinementSurfaces* level $< 4$) rendered the mesh too coarse to capture the interaction between particles and flow inside of the aperture, while larger values come at a much higher computational cost."

**Please specify the boundary conditions.**

We have added the following specifications to the simulation description:

"The boundary conditions for velocity are a flat velocity profile at the inlet; slip conditions

at top, bottom, front, back and outlet surfaces; and a no-slip condition at the object (MASC). Zero gradient pressure fields are applied at all boundaries."

**Please justify why potentially important forces acting on particles in turbulent/shear flows are neglected (lift force, pressure gradient force)**

We have rearranged the sections and wording in the title (as suggested by Reviewer #2) to reflect that the observations analysis is the featured part of our work, and that the relatively simple simulations serve only a supporting role. We have added the following to the conclusions section:

"Relatively simple simulations were carried out here to support the findings of the observations analysis. We used only a single set of particles with limited, yet representative characteristics to support observations analysis with simulated particle responses to MASC-perturbed flow. Future work could include a much more diverse set of particle shapes, sizes, and densities, as well as other forces that have been neglected in this work and a turbulent dispersion model."

**Where are the starting positions of the particles? (could be included e.g. in one of the figures 1(c)-1(e))**

We have expanded the description of initial positions in the text as follows:

"The particles fall downward at an initial velocity of $1\,\mathrm{m\,s^{-1}}$ from an initial height of $3\,\mathrm{m}$ above the MASC in the $-z$ direction under the force of gravity, reaching an average terminal velocity of $1.05\,\mathrm{m\,s^{-1}}$ well before encountering flows perturbed by the MASC. Initial particle positions are ∼2 to 20 m away from the MASC in the upstream horizontal direction, depending on the flow velocity. These initial positions were evaluated to ensure they fell into the center of the aperture."

**2) Furthermore, I would suggest using more particles and particles of different size and shape according to the different riming classes seen in the measurements. And, as the influence of turbulent wind on the collection characteristics**

**is examined, a model for turbulent dispersion should also be included. The finding that fall velocity is reduced with increasing wind speed could maybe have been made without the particle simulation part by just looking at the fluid velocities inside/above the collection area. If so, the particle simulations might as well be left out and more detailed particle simulations would be something for further work.**

Although we neglect some of the details as you point out, we feel that it is still necessary to show the influence of the perturbed flow on the mean fall speed of a set of particles with which to compare the MASC measurements. As stated above in response to #1, we have rearranged the sections and wording in the title (as suggested by Reviewer #2) to reflect that the observations analysis is the featured part of our work, and that the relatively simple simulations serve only a supporting role. We have added the following to the conclusions section:

"Relatively simple simulations were carried out here to support the findings of the observations analysis. We used only a single set of particles with limited, yet representative characteristics to support observations analysis with simulated particle responses to MASC-perturbed flow. Future work could include a much more diverse set of particle shapes, sizes, and densities, as well as other forces that have been neglected in this work and a turbulent dispersion model."

**3) As in principle the dependence of the collection efficiency on particle dynamic properties is investigated, I would suggest to include some thoughts on and references to general particle dynamics. Namely, there is the particle Stokes number, defining the ability of particles to respond to sudden changes in the carrier fluid's velocity. It is proportional to the product of particle density and the square of the diameter. Aggregates with comparatively low densities would have a low Stokes number, indicating that they follow the flow better than more dense aggregates. Additionally, bodies moving through fluids tend to orient in a manner that the drag is maximised. For oblate particles this would mean that their "plane"**

**is oriented perpendicular to the flow direction. With these two principles a lot of the paper's findings can be explained quite well.**

We have added the following to the discussion section (note that Figs. 8 & 6 and Table 1 correspond to Figs. 12 & 10 and Table 3 in the original manuscript, respectively):

"Larger aggregates with negligible riming tend to be more susceptible than smaller more dense particles to disturbance by surface winds and associated turbulence, with a tendency for a more vertical orientation (Fig. 8), slower fall speeds (Fig. 6), and lower frequency of occurrence with higher wind speeds (Table 1) than other riming classes. The Stokes number is defined as the dimensionless ratio of the particle relaxation time to its terminal velocity in still air $v_t/g$, and a characteristic time of isotropic, homogeneous turbulent flow. Snowflakes with low Stokes numbers tend to follow the flow, becoming trapped in the vortices with the orientation aligning with the local velocity gradient (Voth & Soldati, 2017). The implication is that large, low-density, aggregate-type hydrometeors with relatively small values of $v_t$ compared to more heavily rimed particles have low values of the Stokes number and are more likely to follow the motions of any turbulent flow induced by the MASC aperture. This finding is consistent with prior work by Theriault et al. (2012) who showed that for a Geonor gauge inside a single Alter shield, higher-density, faster-falling hydrometeors are collected most efficiently."

Voth, G. A., & Soldati, A. (2017). Anisotropic particles in turbulence. Annual Review of Fluid Mechanics, 49, 249-276. https://doi.org/10.1146/annurev-fluid-010816-060135

**4) Regarding the orientation of particles: As non-spherical particles tend to arrange themselves in some preferential angle to the flow, the concept of determining their orientation seems questionable even for medium wind speeds between 1.5 m/s and 5.0 m/s, as the flow above and inside the measurement region seems to be heavily influenced by the device. What are the authors' views on this?**

This is why we deemed it necessary to show orientations in even lighter winds (i.e., wind speeds as low as $< 0.5\,\mathrm{m\,s}{-1}$ in Fig. 12 from the original manuscript). This figure

shows not only what the preferred orientation angle is such light winds, but also how sensitive the larger, low-density aggregates are.

**Minor remarks:**

**Section 1: L46f: suggested change from "characterize" to "were used to characterize" or similar**

Fixed as suggested

**L58f: Isn't the collection region cylindrical or circular, not ring-shaped?**

This has been changed to a "hollow, decagonal-prism-shaped collection volume" to more accurately describe the shape of the collection region.

**L58ff: I would suggest to include some information about the different parts' positions in Fig. 1a)**

Added information about parts' positions as follows (note that Fig. 11 corresponds to Fig. 1 in the original manuscript):

"The system's casing houses three cameras focused on a point at the center of the collection volume 10 cm away, with each camera separated by 36° (for more details, see Fig. 1 from Garrett et al. (2012). A coupled system of directly opposing near-infrared emitters and detectors (attached to the decagonal prism in Fig. 11(a)), vertically separated by 32 mm, detect falling hydrometeors larger than approximately 0.1 mm in maximum dimension (Garrett & Yuter, 2014). This triggers the cameras and three high-powered LEDs located directly above on top of the casing (see Fig. 11)."

**L70ff: Maybe the number of references can be reduced, where possible?**

We removed all Garrett et al. references that had already been cited and added "e.g.," to show that this is a subset of relevant references.

**L90f: also include influence on riming degree among "fall speed, fall orientation,**

**and size distribution"?**

Added as suggested

**Section 2: Please specify more clearly: It seems that first a steady state simulation with simple-Foam is done, and afterwards particles are tracked through a "frozen" flow field, but it is not actually mentioned.**

The following statement was added to the CFD description: "The flow is allowed to reach steady-state prior to tracking particles through a 'frozen' flow field."

**L96: Which solver? Maybe simpleFoam should be mentioned here already.**

Wording in this paragraph has been rearranged to correct this and mention simpleFoam earlier.

**L100: simpleFoam instead of simpleFOAM**

Corrected

**L115: Please specify the drag coefficient function.**

We have updated the text to state, "The drag coefficient $C_D(Re_p)$ is defined as $C_D = (24/Re_p)(1 + Re_p^{2/3}/6)$ for $Re_p \leq 1000$ and $C_D = 0.44$ for $Re_p > 1000$"

**L115 & L117:** $Re$ **should be** $Re_p$

Corrected

**Fig. 2: I would suggest writing "wind directions" instead of "wind vectors". What crosssection is meant? Maybe the plane for the arrows? Please specify. A more detailed view of the results inside/around the measurement section would be nice, as it would be possible to examine the wind speed there more detailed. Is it possible to show some exemplary particle path lines? (additional images could be included as supplementary material)**

We have updated the caption to state:

"Simulated wind field around the MASC with undisturbed winds set at $1\,\mathrm{m\,s^{-1}}$ towards the positive y–direction. Color represents the vertical wind speed $v_{f_z}$, and arrows show wind directions on the y–z plane. The plane in which the arrows are located is aligned with the center of the aperture on the y–z plane, and x–positive points out of the page."

In response to the request for a figure showing examples of particle trajectories from two reviewers, we will add at least one figure to show this in the revised manuscript. Attached is a preview for a flow of $1\,\mathrm{m\,s^{-1}}$:

**Fig. 3 and Fig. 4: Please indicate where the data is sampled.**

We have added the statement, "Data are sampled at the center of the aperture" to each of these captions.

**L130: "in the k-$\omega$-SST closure model"**

Corrected

**Section 3: L137: "the angle between $D_{max}$ and the local horizontal" should be "the angle between the major axis and the local horizontal"**

Corrected as suggested and also clarified the definition of maximum dimension in the preceding sentence to say that it is defined as the length of the major axis

**L143: Is the average of the three images the best guess for the actual orientation angle? How "stable" is the detection of orientation angles for rimed particles, as they might have no clear major axis?**

We have added to the discussion section:

"The average of the three images is not the best guess for the true orientation angle in all cases. For example, depending on the azimuthal orientation with respect to the central camera, the particle's major axis may not be resolved entirely. However, Jiang

et al. (2019) showed that the azimuthal orientation is correlated with the wind direction, with particles' major axes tending to align with the wind direction. In our case, this would imply that the major axis is often oriented such that it cannot be entirely resolved by any of the three cameras. More work needs to be done to investigate the limitations of the MASC-determined orientation angle."

**Fig. 6(b) might be dropped in my opinion**

We have decided to keep this as no other reviewers commented on it, and we feel it adds some clarification through an additional perspective

**L171: Why were the PDFs adjusted with a Gaussian kernel and how do the original data look?**

Added to the end of the Hydrometeor observations – methods section:

Results are presented here in the form of probability density function (PDF) estimates, calculated using a kernel density estimator of the form

$$\hat{f}(x_0) = \frac{1}{n_s h} \sum_{i=1}^{n_s} K\left(\frac{x_0 - x_i}{h}\right) \tag{1}$$

where $x_0$ is a real value of the distribution being estimated, $x_i$ is a random sample from the distribution, $n_s$ is the sample size, and $h$ is the bandwidth (Wilks, 2011). The Gaussian smoothing function is $K(x) = (2\pi)^{-1/2} \exp(-x^2/2)$ for a random variable $x$, and $h$ is optimized according to Bowman & Azzalini (1997) to produce a smooth curve.

**Table 3: In the regime of $U_{sfc} \leq 5\,m/s$: Are the data for smaller $U_{sfc}$ also contained in values for larger $U_{sfc}$? (e.g. is the data of $U_{sfc} \leq 0.5\,m/s$ also in $U_{sfc} \leq 1.0\,m/s$?)**

Yes and we have added the following clarification to the table's caption: "Less restrictive wind speed categories (e.g., $\leq 5\,\mathrm{m\,s^{-1}}$) include data from more restrictive categories (e.g., $\leq 1.5\,\mathrm{m\,s^{-1}}$)"

**L187f: If something different is simulated, then it should not be compared to that. There seems to be a significant difference between the measurements and the simulations. Please include some discussion about this difference.**

This comparison statement has been removed and the following paragraph has been added to the discussion:

"For unshielded MASC measurements, the simulations demonstrated that the separation of flow leads to an upward flow velocity component above the aperture, which tends to decrease the mean fall speed of particles falling into the aperture. The mean value of the fall speeds measured by the unshielded MASC shown in Fig. 5(b) ($U_{sfc} \leq 5\,\mathrm{m\,s^{-1}}$) is $0.55\,\mathrm{m\,s^{-1}}$, which is similar to the simulated mean of $0.47\,\mathrm{m\,s^{-1}}$ for ambient wind speeds of $5\,\mathrm{m\,s^{-1}}$ in Table 3. However, the mean MASC-measured fall speed for $U_{sfc} \leq 1.5\,\mathrm{m\,s^{-1}}$ from Fig. 5(c) is only $0.60\,\mathrm{m\,s^{-1}}$, whereas the corresponding simulation-determined mean fall speed is $\sim 0.95\,\mathrm{m\,s^{-1}}$. This is a result of the Gaussian distribution of particle fall speeds from the simulations (not shown) not matching the observations, where distributions tend to either heavily favor the low-fall-speed mode or are bi-modal for light winds ($U_{sfc} \leq 1.5\,\mathrm{m\,s^{-1}}$). Therefore, while the primary effect of perturbed winds slowing particle fall speeds is well represented in the simulations, the details appear to be more complicated in reality."

**Fig. 10: The low fall speed mode seems to be less prominent ($U_{sfc} <= 1.0$ and $1.5\,m/s$) or even vanishes ($U_{sfc} <= 0.5\,m/s$) for aggregates. Is there any explanation for this?**

It is not clear to us why this is the case. One idea is that the spectrum of aggregate fall speeds is less susceptible to broadening by turbulent eddies, but this would need to be investigated in a separate work.

**Section 4: As metioned above, please include some discussion of the simulation results and how they compare to the measurements. As mentioned in major remarks, I suggest including some of these thoughts: Falling objects tend to**

**orient in maximum drag conditions, which matches the findings here. (compare L196ff) As aggregates seem to have smaller fall velocities than particles of the other two categories, they should be oriented more vertical to orient in maximum drag conditions when experiencing the same horizontal wind speeds. This again matches with the findings. (compare L201) Aggregates are more prone to get blown away by the surface winds as they have comparatively low Stokes numbers and high surface-to-density-ratios in comparison to more rimed particles. (compare L205ff)**

As mentioned above, we have added the following to the discussion section (note that Figs. 8 & 6 and Table 1 correspond to Figs. 12 & 10 and Table 3 in the original manuscript, respectively):

"Larger aggregates with negligible riming tend to be more susceptible than smaller more dense particles to disturbance by surface winds and associated turbulence, with a tendency for a more vertical orientation (Fig. 8), slower fall speeds (Fig. 6), and lower frequency of occurrence with higher wind speeds (Table 1) than other riming classes. The Stokes number is defined as the dimensionless ratio of the particle relaxation time to its terminal velocity in still air $v_t/g$, and a characteristic time of isotropic, homogeneous turbulent flow. Snowflakes with low Stokes numbers tend to follow the flow, becoming trapped in the vortices with the orientation aligning with the local velocity gradient (Voth & Soldati, 2017). The implication is that large, low-density, aggregate-type hydrometeors with relatively small values of $v_t$ compared to more heavily rimed particles have low values of the Stokes number and are more likely to follow the motions of any turbulent flow induced by the MASC aperture. This finding is consistent with prior work by Theriault et al. (2012) who showed that for a Geonor gauge inside a single Alter shield, higher-density, faster-falling hydrometeors are collected most efficiently."

**L236: "enhanced are reduced" should read "enhanced or reduced"**

Corrected

[Figure]

[Figure]

[Figure]

**Fig. 1.** Simulated particle trajectories for wind speed of 1 m sˆ-1

---

## Author Comment (AC2) · 27 Nov 2020

**The manuscript investigates the aerodynamic impact of the particles' fall speed measured by a Multi-Angle Snowflake Camera (MASC) using field measurements and Computational Fluid Dynamics (CFD) simulations. It compares the fall speed PDF measured by the MASC and the K-band radar located at the same site. The distribution of fall speed differs from the two instruments and the numerical simulations suggested that the fall speed measured in strong winds ($> 5$ m/s) would record slower falling particles when not shielded. Similar results were found using the simulations. Overall, this study helps to improve the quality control**

[Figure]

procedure of the MASC data and contributes significantly to the field of snowfall measurement. It fits well in this journal as it improves the methodology to be used to quality control MASC data. The manuscript is very well written and clear. The figures are also clear and well described in the caption. I have, however, a few main and minor comments should be considered before publication.

**Main comments**

**1. The manuscript gives the impression that the main point is the CFD simulations where it would have been used to study the collection efficiency or to develop transfer function to adjust the MASC measurements. After reading the manuscript, the CFD simulations are used only to explain the field measurements. Given that, I think that the authors should add a methodology section after the introduction that explains the approach taken in this study, which includes the field measurements and the simulations. It may also be useful to present the measurements before showing the results from the simulations.**

We agree and have decided to present observations first in addition to changing the title to "Arctic observations and numerical simulations of surface wind effects on Multi-Angle Snowflake Camera measurements" to make it more clear that simulations are supporting the observations and not the other way around.

**2. More details should be given about the simulations conducted such as, for example, the number and shape of the mesh used.**

We have modified the description as follows:

"The *snappyHexMesh* tool requires an existing base mesh to work with, which is generated from *blockMesh* and is represented in Table 2. For *snappyHexMesh*, two of the most important parameters are *nCellsBetweenLevels*, set to 3, and the *refinementSurfaces* level, which is set to a minimum of 4 and maximum of 5. This brings the total number of cells to 131,864 when the block is 4 m × 4 m × 5 m. These values were

[Figure]

determined through analysis of grid independence. For *blockMesh*, the resolution of 25 cm × 25 cm × 25 cm provided the most efficient mesh for a fixed *snappyHexMesh*. The *snappyHexMesh* parameters were also determined through testing; lower values (e.g., *nCellsBetweenLevels* < 3 or *refinementSurfaces* level < 4) rendered the mesh too coarse to capture the interaction between particles and flow inside of the aperture, while larger values come at a much higher computational cost."

**Did you use the integrated trajectory simulations or developed one?**

We have clarified by adding the following sentence to the description of CFD simulations:

"The integrated, semi-developed *solidParticleFoam* is used to simulate particle trajectories, with gravity included to supplement the developed simulation."

**Could you add a figure that includes examples of particles' trajectories?**

We have added this to the CFD simulations section. Attached is a preview for a flow of $1\,\mathrm{m\,s^{-1}}$.

**In Table 2, only one size of particle was used. For dry snow and aggregates, the fall speed does not change much with diameter according to Rasmussen et al. (1999). However, the fall speed of rimed particles can vary a lot with sizes. Why not use more particles' sizes? What would be the impact on your results? Why did you choose a diameter of 2 mm and not 1 mm?**

We have added the following to the conclusions section:

"Here we used only a single set of particles with simple, yet representative characteristics to support observations analysis with simulated particle responses to MASC-perturbed flow. Future work could include a much more diverse set of particle shapes, sizes, and densities."

**Please also describe in more details the simulations. For example, is there an**

**updraft as found in previous studies (ex: Colli et al. 2016a,b; Theriault et al. 2012) for the Geonor (shielded and not)? How do you explain that slower falling snowflakes fall is detected by the MASC in stronger winds? Add any other details that could help better understanding the results from the simulations.**

We have modified and added the following statements to the CFD simulations text describing the simulated flow perturbed by the MASC as shown in Fig. 2 (which is now Fig. 12 in the revised manuscript):

"There is a clear separation of flow at the upstream side of the aperture, a relatively large upward component above the aperture at the upstream side, and a smaller downward component within the aperture. The fall speeds of particles carried into the aperture by the prevailing flow are decreased by this upward component of the flow, which increases with increasing wind speeds."

**3. The simulation as well as the measurement shows that an unshielded MASC leads to a decrease of the fall speed. Can you add a brief explanation in the manuscript? It seems counter-intuitive as faster falling particles would tend to fall in the gauge in stronger winds.**

In addition to the above statements added to the CFD simulations section, we have also added the following sentence to the conclusions:

"The simulations revealed that an upward component of perturbed flow at the upstream side of the MASC aperture increases in magnitude with increased wind speeds, and that this leads to decreasing mean particle fall speeds with increased horizontal wind speeds."

**4. At lower wind speed, larger aggregates tend to be more detected by the MASC than at higher wind speeds. In theory larger ones would fall faster and would not be deflected. How do you explain this finding? Could it be because larger aggregates in strong winds would breakup? Or is it common to report large ag-**

**gregates in windy conditions at that site? Did you compare with the climatology of solid precipitation at that location?**

We have added the following to the discussion section (note that Figs. 8 & 6 and Table 1 correspond to Figs. 12 & 10 and Table 3 in the original manuscript, respectively):

"Larger aggregates with negligible riming tend to be more susceptible than smaller, more dense particles to disturbance by surface winds and associated turbulence, with a tendency for a more vertical orientation (Fig. 8), slower fall speeds (Fig. 6), and lower frequency of occurrence with higher wind speeds (Table 1) than other riming classes. The Stokes number is defined as the dimensionless ratio of the particle relaxation time to its terminal velocity in still air $v_t/g$, and a characteristic time of isotropic, homogeneous turbulent flow. Snowflakes with low Stokes numbers tend to follow the flow, becoming trapped in the vortices with the orientation aligning with the local velocity gradient (Voth & Soldati, 2017). The implication is that large, low-density, aggregate-type hydrometeors with relatively small values of $v_t$ compared to more heavily rimed particles have low values of the Stokes number and are more likely to follow the motions of any turbulent flow induced by the MASC aperture. This finding is consistent with prior work by Theriault et al. (2012) who showed that for a Geonor gauge inside a single Alter shield, higher-density, faster-falling hydrometeors are collected most efficiently."

Voth, G. A., & Soldati, A. (2017). Anisotropic particles in turbulence. Annual Review of Fluid Mechanics, 49, 249-276. https://doi.org/10.1146/annurev-fluid-010816-060135

**Some minor comments:**

**1. Lines 5-7: This sentence mentions that the simulations are compared with observations. However, I understood that in this study that the catch efficiency of the instruments is not computed from the simulations and compared with the measured one. But the simulations are used to explain the decrease in fall speed measured in strong winds. This is related to major comment #1. It should be rephrased for clarity.**

This has been reworded to state, "Here we present analysis of Arctic field observations with and without a Belfort double Alter shield and compare the results to computational fluid dynamics (CFD) simulations of the airflow and corresponding particle trajectories around the unshielded MASC."

**2. Lines 45: Newman et al. (2009) also conducted CFD simulations in the vicinity of a snowflake video imager. Should probably add the paper to this paragraph. Newman, A. J., P. A. Kucera, and L. F. Bliven, 2009: Presenting the Snowflake Video Imager (SVI). J. Atmos. Oceanic Technol., 26, 167–179, https://doi.org/10.1175/2008JTECHA1148.1.**

Added to the end of that paragraph: "CFD simulations were also analyzed for wind flow along the optical axis of a snowflake video imager, with eddies dissipating approximately 1 m downstream of the camera housing and only minor modifications to the wind field (Newman et al., 2009)."

**3. Figures 7, 9, 11 and 13: Those figures compare data taken with an unshielded and a shielded MASC. Please clarify in the caption that the shielded and unshielded data were collected during two different periods.**

Added to each of those figures' captions: "Unshielded and shielded MASC observations are from two separate periods: 29 November 2015 to 21 August 2016 and 22 August 2016 to 28 August 2018, respectively."

**4. Lines 185: Usfc is defined. Could you explain it further and how it compares with standard measurements of wind speed at the instrument's height (or at 10 m)? I may have missed the explanation in the text.**

We added the following clarifying sentence to the methods section: "The wind measurement is taken at a standard height of 10 m, which is estimated to be 5(9) m higher than the unshielded(shielded) MASC shown in Fig. 1(2)," where Figs. 1 & 2 correspond to Figs. 5 & 6 in the original manuscript.

[Figure]

**5. Lines 189-194: The authors forgot to introduce figure 10. Only Figure 10c is referred to.**

We have modified this sentence that begins on line 192 in the original manuscript: "When separated by riming class (Fig. 6), shielded MASC fall speed distributions show discernible differences only for the lightest winds," where Fig. 6 corresponds to Fig. 10 in the original manuscript.

**6. Lines 200-201: Please clarify that sentence. I don't understand what you mean by 'more vertical'?**

We have clarified the meaning in the sentence as follows:

"...shielded MASC orientation angles tend to be larger for sparsely-rimed aggregates (Fig. 12), meaning their major axes are less frequently oriented within the horizontal plane."

**7. Lines 224-229: For clarity, the authors could remind the reader that larger aggregates fall slower than the rimed particles as in Figure 10.** We have modified the sentence as follows:

"Larger aggregates with negligible riming tend to be more susceptible than smaller, more dense particles to disturbance by surface winds and associated turbulence, with a tendency for a more vertical orientation (Fig. 8), slower fall speeds (Fig. 6), and lower frequency of occurrence with higher wind speeds (Table 1) than other riming classes."

Where Figs. 6 & 8 and Table 1 correspond to Figs. 10 & 12 and Table 3 in the original manuscript, respectively.
* * *
[Figure]

[Figure]

**Fig. 1.** Simulated particle trajectories for wind speed of 1 m sˆ-1

---

## Author Comment (AC3) · 27 Nov 2020

**This manuscript summarizes work focused on snow particle observations from a MASC deployed at the NSA site. Specifically, the authors investigate the impact of wind shielding on the MASC instrumentation both from field measurements and from Computational Fluid Dynamics (CFD) simulations. The authors find that the fall speeds from the MASC agreed much better with Doppler velocities from a co-located KAZR for the wind shielded events. Additionally, the CFD simulations indicated slower particle fall speeds when the MASC was unshielded. In general, I think this work is novel and advances the field of in situ snow parti-**

[Figure]

cle observations. I would also like to commend the authors on an exceptionally well-written manuscript, which had a clear narrative and was enjoyable to read.

I have one major comment and a few minor that should be addressed prior to publication:

Major Comment:

It is unclear to me how many distinct events were used to comprise the observations that were presented. In the methods, the timelines of the MASC deployment unshielded (Feb 2015 – Aug 2016) and shielded (Aug 2016 – Aug 2018) are outlined (33 months total), however it is not discussed anywhere how many independent events are used in this work. This is key information that is missing from this manuscript as it lends weight to the differences seen between unshielded and shielded observations. This is especially true for Table 3 – as the observations are further divided into wind speed bins and by particle type. A single event could produce 1000s of particle images, so it should be made clear how many independent events were used. This should be added to the methods section – ideally as a table (dates, times). Currently, the manuscript reads as if there are enough observations to say that these fractions of different particle types (in Table 3) are due primarily to the wind shielding impacts, however if there is a low number of independent events (or a low number in a represented wind speed range), then some of these particle type ratios could be from different synoptic or thermodynamic forcing. In addition to including the number of events, the authors should also examine the statistical significance of these differences in particle type (rimed, MR, agg.) for the various wind speed bins (if the N of individual events is large enough).

If only a few independent events were used in this work, I think this should be made clear and the language should reflect that is the case. The implication in the paper (whether purposeful or unconscious) is that the differences in parti-

**cle type ratios seen in shielded versus unshielded at various wind speeds are a product purely from mitigating the wind to the MASC. However, if very few independent snow events were used in these comparisons the synoptic and thermodynamic conditions could be influencing the ratios of rimed, MR, and aggregate particles.**

We have added a paragraph (rather than a table as suggested due to the large number of events) to Section 3.1 (Hydrometeor observations – methods):

"A total of 158,057 particles from 266 distinct events are included here, with 51 events from the unshielded period of 29 November 2015 to 21 August 2016, and 215 events from the shielded period of 22 August 2016 to 28 August 2018. Distinct events were identified by a length of time between MASC precipitation measurements of >12 hours, or by a length of time of >3 hours with an accompanying change of pressure of at least 2 mb. These thresholds were determined by analyzing the period of 4 to 17 December 2017, during which 14,528 precipitation particles were associated with five distinct events as determined by manual inspection of the KAZR reflectivity time series (not shown). Differences in riming class composition for various wind speed categories are determined to be statistically significant by comparing $\chi$ distributions using the two-sample Kolmogorov-Smirnov Test at a 5% significance level. In each test, one sample is from the high-wind category ($U_{sfc} > 5\,\mathrm{m\,s^{-1}}$) and the other is from one of the chosen low-wind categories."

The number of distinct events for each case in Table 3 has been added as a whole number in parentheses, and statistical significance is indicated with * (for wind-shielded cases only). A screen capture of the updated Table 3 is attached.

**Minor Comments:**

**Figure 2 illustrates a CFD simulation across the MASC in the +y direction, which is roughly parallel to the cameras that protrude above the opening. And Fig. 3 shows the impact of the fall speeds for ambient winds in both +y and –x (which I**

**read to be winds toward the cameras). What is the impact of the winds originating from behind the cameras, as this is a large obstacle adjacent to the observing ring? I assume that this direction (+x) will have a larger impact on the particle fall speeds (if I am reading the orientation of the axes correctly). Did you do simulations with the wind originating from behind the cameras?**

We added the following sentence towards the end of the CFD Simulations section:

"Although there is little difference between the wind directions shown, particles carried by wind blowing in the +x direction were almost entirely blocked by the LEDs located on top of the MASC, especially for speeds of $> 2\,\mathrm{m\,s^{-1}}$ (not shown)."

**Along those same lines, wind direction impacts were noted in the discussion about the simulations (minimal), but not in the observations. Was there any analysis on the impacts of wind direction from the observational perspective?**

Only a limited amount of wind direction analysis was performed to understand prevailing wind directions. However, we feel that a proper analysis could not be performed for this site without a better understanding of the precision of MASC alignment and how the wind direction changes between the 10-m wind measurement height and the MASC height ($\sim$5 m when unshielded, $\sim$1 m when shielded). Furthermore, the LEDs on top of the MASC affect a relatively limited range of wind directions, and actual particle trajectories are much more complicated than our simulations, making wind direction analysis much more complicated than that of wind speed.

**The MASC fall speeds were compared to the KAZR Doppler velocities, and it seems that the mean Doppler velocity from the cloud base to near-surface (I assume) profile was used – is that correct? If so, what was the lowest near-surface bin used in the Doppler velocity profile mean calculation (assuming near-surface to cloud base mean value)? Also, the snow particles can change between the cloud base and the surface – so what is the advantage of using the mean DV value of the profile (near-surface to CB) versus simply using the near-surface**

**Doppler velocity? My instinct is that using the near-surface Doppler velocity value would give you a more direct comparison to the MASC**

The intent in using the mean value for all height bins below cloud base was to average out any errors that might be prone to any one height bin. In any case, a comparison of results using the lowest bin vs. all bins below cloud base revealed no substantial differences in the KAZR mean Doppler velocities.
* * *
[Figure]

| Category | $U_{sfc}$ | | | | |
|---|---|---|---|---|---|
| | $> 5\,\mathrm{m\,s^{-1}}$ | $\leq 5\,\mathrm{m\,s^{-1}}$ | $\leq 1.5\,\mathrm{m\,s^{-1}}$ | $\leq 1.0\,\mathrm{m\,s^{-1}}$ | $\leq 0.5\,\mathrm{m\,s^{-1}}$ |
| **No Wind Shield** | **2,249 (27)** | **5,097 (31)** | **460 (9)** | **167 (7)** | **32 (4)** |
| Aggregates | 176 (8%,16) | 1,522 (30%,22) | 67 (15%,6) | 15 (9%,5) | 5 (16%,2) |
| Moderately Rimed | 1,209 (54%,25) | 2,891 (57%,27) | 315 (68%,8) | 115 (69%,6) | 14 (44%,4) |
| Rimed | 864 (38%,13) | 684 (13%,19) | 78 (17%,5) | 37 (22%,2) | 13 (41%,2) |
| **Wind Shield** | **85,151 (181)** | **58,939\* (140)** | **5,730\* (45)** | **1,372\* (30)** | **161\* (13)** |
| Aggregates | 15,320 (18%,132) | 11,304 (19%,101) | 1,299 (23%,30) | 302\* (22%,21) | 41\* (25%,8) |
| Moderately Rimed | 47,147 (55%,165) | 35,820\* (61%,128) | 3,477\* (61%,38) | 855\* (62%,26) | 86 (53%,12) |
| Rimed | 22,684 (27%,151) | 11,815\* (20%,107) | 954\* (17%,35) | 215\* (16%,21) | 34 (21%,6) |

**Fig. 1.** Table 3, updated

---

## Author Response (AR2)

**Responses to Additional Minor Comments for "Arctic observations and numerical simulations of surface wind effects on Multi-Angle Snowflake Camera measurements"**

Kyle E. Fitch[1,2], Chaoxun Hang[3,4], Ahmad Talaei[1], and Timothy J. Garrett[1]

[1]Department of Atmospheric Sciences, University of Utah, Salt Lake City, 84112, USA
[2]Department of Engineering Physics, Air Force Institute of Technology, Wright-Patterson Air Force Base, Ohio, 45433, USA
[3]Department of Civil Engineering, Monash University, Clayton, 3168, Australia
[4]School of Oceanography, Shanghai Jiao Tong University, Shanghai, 200240, China

**Correspondence:** Kyle Fitch (kyle.fitch@afit.edu)

**1  Reviewer #1 Minor Comments**

**L101: Should "distinguished" read "distinguish"?**

Yes, corrected - thanks

**L212ff: How many cells are there after running snappyHexMesh?**

5    This is mentioned in lines 214-215: "This brings the total number of cells to 131,864 when the block is 4 m × 4 m × 5 m."

**L232ff: If you have one inlet and slip-/no-slip-conditions everywhere else for velocity, an incompressible flow is not possible. Are the authors sure that the boundary conditions given here are correct?**

The outlet boundary condition was omitted by mistake. We have added the statement, "The 'inletOutlet' outlet boundary condition was used, which provides a generic outflow condition," to this paragraph.

10  ## 2  Additional Minor Comment

**L138: I'm not sure I understand the methodology correctly. Apparently, the authors used mean Doppler velocity (MDV) as a proxy for the fall velocity? Different than the spectrum width and the edges of the Doppler peak (which were used in Matt Shupe's paper if I remember correctly), MDV is not significantly biased by broadening (in particular below cloud base), so I'm not sure why you mention the Shupe paper here. However, radar return is always dominated by the**

15  **big particles which is why MDV might be biased towards higher values in comparison to a 'real' mean fall velocity. I would be careful in referring to MDV as 'ground truth' or calling it 'fall velocity', because the comparison of MDV to MASC fall speed is like comparing apples to oranges. The language should reflect that so I would recommend to the KAZR data as mean Doppler velocity.**

Yes, MDV was used as a proxy for the fall velocity. We removed the reference to broadening in Shupe's work since it is not
20  relevant here, as you pointed out. Rather than referring to MDV as a ground truth fall speed, the first sentence in this paragraph

was changed to: "For comparison to MASC fall speeds, mean Doppler velocity was calculated from the volume of scattering hydrometeors detected by a co-located Ka-band ARM Zenith-pointing Radar (KAZR)." Furthermore, all references to KAZR fall speed were changed to KAZR mean Doppler velocity (throughout the paper, including captions).